# Pathway-specific responses of isoprene-derived secondary organic aerosol formation to anthropogenic emission reductions in a megacity in eastern China

Huilin Hu[1], Yunyi Liang[2], Ting Li[2], Yongliang She[2], Yao Wang[1,3], Ting Yang[1], Min Zhou[4], Ziyue Li[1], Chenxi Li[1], Huayun Xiao[1], Jianlin Hu[2], Jingyi Li[2,*], Yue Zhao[1,*]

[1]School of Environmental Science and Engineering, Shanghai Jiao Tong University, Shanghai 200240, China

[2]Jiangsu Key Laboratory of Atmospheric Environment Monitoring and Pollution Control, Collaborative Innovation Center of Atmospheric Environment and Equipment Technology, School of Environmental Science and Engineering, Nanjing University of Information Science and Technology, Nanjing 210044, China

[3]Ningbo Ecological and Environmental Monitoring Center, Ningbo 315012, China,

[4]State Environmental Protection Key Laboratory of the Cause and Prevention of Urban Air Pollution Complex, Shanghai Academy of Environmental Sciences, Shanghai 200233, China

Correspondence: Yue Zhao (yuezhao20@sjtu.edu.cn); Jingyi Li (Jingyili@nuist.edu.cn)

**ABSTRACT**:Isoprene-derived secondary organic aerosol (iSOA) represents a major biogenic source of atmospheric OA, with its formation strongly influenced by anthropogenic emissions. However, long-term iSOA measurements in polluted urban regions remain scarce, limiting the understanding of anthropogenic influences on iSOA formation. In this study, field observations of iSOA were conducted in Shanghai, China during summers and winters of 2015, 2019, and 2021, aiming to assess the iSOA response to emission reductions over this period. The particulate iSOA tracers, formed via reactive uptake of isoprene-derived epoxides, were measured by gas chromatography-mass spectrometry and high-resolution liquid chromatography-mass spectrometry. After accounting for the measurement uncertainties, both total iSOA and organosulfates (OSs) tracers (including 2-methyltetrol sulfates and 2-methylglyceric acid sulfate) decreased annually, while summertime polyol tracers like 2-methyltetrols (2-MTs) and 2-methylglyceric acid (2-MG) varied insignificantly despite strong NOx reductions. Declining aerosol reactivity toward isoprene-derived epoxides and reduced atmospheric oxidizing capacity drove the decrease in OSs but could not explain the trend of summertime polyols. Simulations with the Community Multiscale Air Quality model in 2015 and 2019 captured the decrease in total iSOA and OSs, confirming a driving role of chemical processes. However, the model failed to replicate relatively stable polyol levels in summer, suggesting additional factors (e.g., potential unaccounted sources of 2-MTs and methacrolein, the precursor to 2-MG) may buffer their variations. These findings highlight pathway-specific iSOA responses to emission reductions in a megacity and the importance of targeted anthropogenic emission reductions for mitigating biogenic SOA formation through regulating atmospheric oxidizing capacity and aerosol reactivity.

## 1. Introduction

Isoprene (2-methyl-1,3-butadiene, $C_5H_8$), mainly emitted by terrestrial vegetation, represents the largest source of atmospheric non-methane hydrocarbons, with a global emission flux of 500-750 Tg $yr^{-1}$ (Guenther et al., 2006). Because of its large abundance and high reactivity, the oxidation of isoprene contributes significantly to the formation of secondary organic aerosol (SOA) in the troposphere, estimated to 19.6 TgC $yr^{-1}$ (Kelly et al., 2018). The formation chemistry of isoprene-derived SOA (iSOA) has been extensively studied in laboratory, field, and modelling studies, which have shown that anthropogenic pollutants such as nitrogen oxides ($NO_x$) and sulfur dioxide ($SO_2$, the precursor to particulate sulfate) play a crucial role in the formation of iSOA (Shrivastava et al., 2019).

Photooxidation is the dominant fate of isoprene in the atmosphere, during which the C=C bond of isoprene is initially attacked by OH radicals to produce an alkyl radical (Teng et al., 2017), followed by the formation of isoprene hydroxyperoxy radicals ($ISOPO_2$) via oxygen addition (Surratt et al., 2010). The fate of $ISOPO_2$ radicals depends on the environmental conditions. When the $NO_x$ level is sufficiently low (referred to as $HO_2$-dominated conditions), $ISOPO_2$ would mainly react with $HO_2$ radicals to form hydroxyhydroperoxides (ISOPOOH) (Paulot et al., 2009). Further OH radical addition to ISOPOOH produces isomeric isoprene epoxydiols (IEPOX, trans-β-IEPOX: cis-β-IEPOX = 0.63:0.37) with a yield of ~70-80% (Paulot et al., 2009; Bates et al., 2014; St. Clair et al., 2016). The gaseous IEPOX can be taken up into aqueous aerosol and undergo acid-catalyzed reactions to form polyols, organosulfates (OSs), and oligomers (Hettiyadura et al., 2019; Lin et al., 2012; Surratt et al., 2010). Several constituents including 2-methyltetrols (2-MTs) and their oligomers, $C_5$-alkene triols, methyl tetrahydrofurans, and 2-methyltetrol sulfates (2-MT-OS), which were extensively measured in both laboratory-generated and ambient SOA (Xu et al., 2015; Isaacman-Vanwertz et al., 2016; Surratt et al., 2010; Surratt et al., 2007; Worton et al., 2013; Yee et al., 2020), have been employed as molecular tracers of SOA derived from reactive uptake of IEPOX (IEPOX-SOA).

When the $NO_x$ level is high (referred to as NOx-dominated conditions), $ISOPO_2$ would preferentially react with NO, producing unsaturated carbonyls such as methacrolein (MACR) and methyl vinyl ketone (MVK). A large fraction (~45%) of MACR can be further oxidized by OH radicals via aldehydic H-abstraction, followed by oxygen addition to form an acylperoxy radical, $MACRO_2$

(Orlando et al., 1999). Subsequent reaction between $MACRO_2$ and $NO_2$ yields MPAN, which was
proposed to form hydroxymethyl methyl-α-lactone (HMML) or methacrylic acid epoxide (MAE) (Lin
et al., 2013b; Nguyen et al., 2015). Both of these two epoxides can undergo heterogeneous acid-
catalyzed reactions to form 2-methylglyceric acid (2-MG) as well as its oligoesters and sulfated
derivatives (2-methylglyceric acid sulfate, 2-MG-OS), which serve as the molecular tracers of
HMML&MAE-derived SOA (Kjaergaard et al., 2012; Lin et al., 2013b).
In addition to altering the fate of $RO_2$ (including $ISOPO_2$ and $MACRO_2$), $NO_x$ also plays an
essential role in regulating atmospheric oxidizing capacity (e.g., the OH level), which governs the
photooxidation rate of isoprene in the atmosphere. Compared to NOx that strongly affect the gas-phase
oxidation chemistry of isoprene, sulfate aerosol mainly exerts influences on the multiphase chemistry
of isoprene-derived epoxides. On one hand, sulfate serves as an important nucleophile in the reactive
uptake of isoprene epoxides to form OSs, which competes with the reaction of epoxides with water (Lin
et al., 2013b). On the other hand, sulfate can alter the physicochemical properties of particles such as
liquid water content (LWC), acidity, and phase state (Zhao et al., 2018; Zhao et al., 2019; Ye et al.,
2018), which are the key factors influencing the multiphase reactivity of epoxides and the formation of
iSOA.
Because of the complexity in the gas-phase and multiphase chemistry leading to iSOA formation,
the product distribution of $HO_2$- versus $NO_x$-dominated pathways as well as their dependence on
different influencing factors (i.e., NOx, sulfate, LWC, and aerosol acidity) varies significantly in
different environments. A number of field measurements have found that the formation of IEPOX-SOA
tracers was predominated over the HMML&MAE-SOA tracers, even in some $NO_x$-rich urban areas
(Zhang et al., 2022; Hu et al., 2008; Yee et al., 2020), while a few studies showed that the production
of HMML&MAE-SOA was more prevalent, e.g., in Tianjin, China (Fan et al., 2020) and urban areas
of California (Lewandowski et al., 2013). Sulfate was found to be the primary driver of isoprene-derived
SOA in rural and urban regions as the iSOA correlated strongly with sulfate while such correlation did
not occur with LWC and pH, e.g., in central Amazonia and southeastern U.S (Xu et al., 2015; Yee et al.,
2020). It should be noted that significant uncertainties exist in the quantification of iSOA tracers in
ambient aerosols due to the lack of authentic standards and/or the presence of measurement artifacts
(e.g., matrix effects) (Fu et al., 2012; Wang et al., 2008; Ding et al., 2014; Fu et al., 2016; Fu et al.,
2014; Fan et al., 2020; Zhong et al., 2021). Furthermore, while the iSOA polyol tracers are formed in
the particle phase, they can actively partition between gas and particle phases due to their semi-volatile
characteristics (Fan et al., 2020; Isaacman-Vanwertz et al., 2016; Nguyen et al., 2015). As a result,
considering their particle-phase concentration only may bias our understanding of the atmospheric
abundance and chemistry of iSOA.
In recent years, great changes have taken place in air pollutant emissions, atmospheric composition,
and air quality in China due to the implementation of stringent clean air policies nationwide. For
example, $PM_{2.5}$ mass concentrations dropped significantly, with a notable reduction in inorganic water-
soluble ions such as sulfate, ammonium, and cations (Liu et al., 2023a; Zheng et al., 2018; Liu et al.,
2023b; Geng et al., 2024), and a relatively small decrease in organic aerosol in eastern China in the past
decade (Liu et al., 2023a; Yao et al., 2023). The reduction of inorganic ions including sulfate and non-
volatile cations ($Na^+$, $K^+$, $Ca^{2+}$ and $Mg^{2+}$) resulted in a significant decrease in LWC and a slight increase
in aerosol acidity in this region (Zhou et al., 2022). However, concentrations of VOCs and ozone
exhibited different trends, with a slight decrease for the former while the latter showing an upward trend
before 2018 and a slight declining trend after 2018 (Wang et al., 2022; Liu et al., 2023a; Yao et al., 2023;
Gong et al., 2025). The changing atmospheric conditions can affect both gas-phase and particle-phase
chemistry, as well as the gas-particle partitioning behavior of organics. However, how the formation of
biogenic SOA, such as iSOA, respond to the changes in anthropogenic emissions and air pollution
conditions in polluted regions remains poorly understood.
The aim of this study is to understand the atmospheric abundance and formation mechanisms of
iSOA under the influence of continuous anthropogenic emission reductions in eastern China. Ambient
$PM_{2.5}$ samples were collected at an urban site in megacity Shanghai in winter and summer of 2015,
2019, and 2021. The concentrations of isoprene-derived polyols and OSs from both $HO_2$- and $NO_x$-
dominated pathways in $PM_{2.5}$ samples were measured using gas chromatography-mass spectrometry
(GC-MS) and high-resolution liquid chromatography-mass spectrometry (LC-MS), respectively, with
the aid of a suite of authentic and surrogate standards. The quantification errors of iSOA tracers were
evaluated and their gas-particle partitioning behaviors were considered. The relative distribution of
measured iSOA tracers from different pathways as well as their inter-annual and seasonal variations
were analyzed and compared to the calculated reaction efficiency of different pathways as well as the
simulated values using the Community Multiscale Air Quality (CMAQ) model, which helps to constrain
the formation mechanisms of iSOA and the key factors driving the inter-annual variation of iSOA tracers.
**2. Materials and methods**
**2.1 Sampling and Chemical Analysis**

Ambient $PM_{2.5}$ samples were collected on the rooftop of a 20-m tall teaching building at Xuhui

Campus of Shanghai Jiao Tong University (31.201° N, 121.429° E) located in the urban center of
Shanghai, China. The site is impacted by a mixed commercial and residential area. The sampling
campaigns were conducted in summer (July 14 to August 9 in 2015; 14 July to 9 August 2019; 7 July
to 5 August 2021) and winter (6 January to 26 January 2016; 27 December 2018 to 16 January 2019;
22 December 2021 to 18 January 2022) of 2015, 2019, and 2021. Sampling started at 8:00 am local
time and lasted for 23 hours. A total of 138 $PM_{2.5}$ samples were collected on prebaked (550 ℃, 6 h)
quartz filters (18 × 23 $cm^2$, Whatman) using a high-volume sampler (HiVol 3000, Ecotech) at a flow
rate of 67.8 $m^3\ h^{-1}$. All sampled filters were wrapped with prebaked aluminum foil and stored at -20 ℃
until analysis.

iSOA polyol tracers, included 2-MG, cis-2-methyl-1,3,4-trihydroxy-1-butene, trans-2-methyl-

1,3,4-trihydroxy-1-butene, and 3-methyl-2,3,4-trihydroxy-1-butene (cis-2-MTB, trans-2-MTB, and 3-
MTB, collectively named as $C_5$-alkene triols), as well as 2-methylthreitol and 2-methylerythritol
(collectively named as 2-MTs), were analyzed by a GC-MS. The details regarding the sample
preparation and analysis protocol are presented in Section S1 in the supplement. An example of the total
ion chromatograms (TIC) of iSOA polyol tracers is shown in Figure S1d. For iSOA polyol tracers,
surrogate standards including erythritol, ketopinic acid, and glycerol were used for quantifications in a
number of studies (Kang et al., 2018; Ding et al., 2014; Zhang et al., 2019; Fan et al., 2020). In this
study, concentrations of 2-MG and 2-methylerythritol were quantified using their authentic standards
(Toronto Research Chemicals, 99.8%), and $C_5$-alkene triols and 2-methylthreitol were quantified by 2-
methylerythritol. The uncertainty for $C_5$-alkene triols was estimated previously and their concentration
was found to be underestimated by 65% when 2-methylerythritol was used as the surrogate
standard(Frauenheim et al., 2022). Furthermore, the uncertainty of 2-methylthreitol quantified by 2-
methylerythritol was assumed to be negligible as the differences in the TIC response for homologues
are determined by carbon number, functional groups, and number of active hydrogen atoms that would
silylate (Stone et al., 2012).
The particulate iSOA OSs were analyzed by a LC-MS employing a reversed-phase column ($C_{18}$,
2.1 mm ×100 mm, 1.7 μm, Waters). The sample extraction procedure and analysis protocol were
described in detail in our previous work (Wang et al., 2021). A brief summary is given in Section S2
and examples of total and extracted ion chromatograms of iSOA OSs are provided in Figure S1a-c. The
lactic acid sulfate (LAS) was used as a surrogate standard to quantify the 2-MT-OS and 2-MG-OS.
Use of surrogate standards would lead to uncertainties in measured concentrations of 2-MT-OS and 2-
MG-OS (Bryant et al., 2020; Bryant et al., 2021), but not alter the inter-annual trend of iSOA OSs.
**2.2 Additional measurements and models**
The concentrations of organic carbon (OC) and element carbon (EC) in filter samples were
measured using a thermal–optical multiwavelength carbon analyzer (DRI, Model 2015). The
concentration of organic mass (OM) was estimated by multiplying the OC by 1.6 (Tao et al., 2014). An
ion chromatograph (Metrohm MIC) was employed to determine water-soluble inorganic compounds
(e.g., sulfate, nitrate, chloride, ammonium, potassium ion, and calcium ion). Temperature, relative
humidity (RH), as well as the concentrations of trace gases and $PM_{2.5}$ were measured at a state-
controlled air quality monitoring station on the Xuhui Campus of Shanghai Normal University, which
is surrounded by residential areas and commercial districts and 4.5 km southwest of the $PM_{2.5}$ sampling
site of this work. The mean values of these meteorological parameters and pollutant concentrations are
listed in Table S1.
LWC, pH, and bisulfate concentration in aqueous aerosols were predicted by the ISORROPIA-II
thermodynamic model (Wang et al., 2021). The molar concentrations of particulate water-soluble
inorganic ions (including sulfate, nitrate, chloride, ammonium, potassium ion, and calcium ion),
temperature, and RH were input into the model, which was run in the forward mode for metastable
aerosols. As the predicted aerosol pH could be underestimated when using particle-phase concentrations
of ion species as inputs only (Hennigan et al., 2015), we adopted the pH values of 2015 and 2019 at a
nearby site reported by Zhou et al. (2022), who used both particulate inorganic ion concentrations and
gaseous ammonia as inputs of ISORROPIA-II. Additionally, the 2021 pH was predicted by the
ISORROPIA-II using particle-phase-only concentrations of ions as input due to lack of gas-phase $NH_3$
data. Previous studies have found that lacking gas-phase inputs of ammonia could lead to under-
prediction of pH using thermodynamic equilibrium models, such as ISORROPIA and E-AIM (Guo et
al., 2015; Song et al., 2018; Hennigan et al., 2015). Guo et al. (2015) found that the pH values were
underestimated by 1 unit on average in southeast US when using only aerosol ammonium data as inputs
in ISORROPIA model. Similarly, Song et al. (2018) found that a 10-fold increase in gas-phase $NH_3$
concentrations roughly corresponds to a 1 unit increase in pH in the ammonia-rich atmosphere like
Beijing. In addition, we found that aerosol pH in 2015 and 2019 predicted using aerosol ammonium
only as input in our study was on average 1 unit lower than that predicted using gas-plus-particle-phase
ammonia as input in Zhou et al. (2022). Thus, we inferred that the lack of gas-phase concentrations of
ammonia might lead to underestimation of pH by ~1 unit in the present study and increased the output
pH estimated using aerosol ammonium only as input by one unit to represent aerosol acidity in 2021.
In addition, considering that iSOA polyol tracers are semi-volatile and water-soluble, which can
partition into OM or dissolve into aerosol liquid water, their gas-phase and particle-phase fractions were
estimated by accounting for the gas-organic phase partitioning using an organic absorptive equilibrium
partitioning model and the gas-aqueous phase partitioning using the Henry's Law. The details of the
estimation method are described in Section S3.

## 2.3 Quality control and quality assurance

Procedural blanks (run every 6 samples) and field blanks (two for each season) were extracted and
analyzed in the same manner as the $PM_{2.5}$ filter samples. Target compounds were found below the
detection limit in the blanks. The recovery of iSOA polyol tracers as represented by that of the internal
standard (ketopinic acid) during GC-MS analysis was determined to be 85±17% and the recovery of
iSOA OSs during LC-MS analysis was 72.5%, as determined using LAS spiked onto the filter in our
previous study (Wang et al., 2021).
As the $PM_{2.5}$ sample extracts contain thousands of multifunctional compounds, there might exist
a significant matrix effect in the analysis of iSOA tracers. The matrix effect of polyol tracers was
evaluated by comparing the signal responses of authentic standards of 2-methylerythritol and 2-MG in
$PM_{2.5}$ extracts to those in pure solvent. Six filter samples representing low and high $PM_{2.5}$ concentrations
in summer and winter were used to test the matrix effect. 16 μL or 160 μL of 2-methylerythritol and 2-
MG standard mixtures (10 ppm) was added to the filter extracts to represent the low and high
concentrations of 2-methylerythritol and 2-MG in the samples. The filters were subsequently analyzed
as described in Section 2.1. The matrix factors of the standards were calculated as the ratio of their
signal response in sample extracts (the signal response of these species in non-spiked extracts was
subtracted) to their signal response in the pure solvent. As shown in Table S2, 2-MG had a matrix factor
of 0.77 ± 0.12, while that of 2-methylerythritol was 1.24 ± 0.08, indicating that the concentrations of 2-
MG were likely underestimated by 23% whereas 2-methylerythritol was overestimated by 24% due to
the matrix effect.

Previous studies have demonstrated that the concentrations of 2-MT-OS and 2-MG-OS were

significantly underestimated due to matrix effect by using reversed phase liquid chromatography-mass
spectrometry (RPLC-MS) (Hettiyadura et al., 2015; Bryant et al., 2020; Bryant et al., 2021). In the
present work, because of lack of authentic standards of isoprene-derived OSs, we are not able to
quantify the absolute value of underestimation in the concentration of 2-MT-OS and 2-MG-OS due to
the matrix effect. However, using ambient $PM_{2.5}$ samples with different concentrations, we can quantify
the relative extent of underestimation in OS concentrations due to matrix effect in different samples,
which allows for an evaluation of uncertainties in the abundance, trend, and relative ratios of different
iSOA tracers in this study. In the matrix effect experiments, the extracts of ambient $PM_{2.5}$ samples with
different concentrations were mixed and the measured signals of 2-MT-OS and 2-MG-OS in mixed
extracts were compared to the sum of OS signals detected separately in individual extracts. The
concentrations of $PM_{2.5}$, sulfate, as well as 2-MT-OS and 2-MG-OS in ambient samples used for this
evaluation are listed in Table S3. The relative matrix effect factor ($F_{matrix}$), defined as the ratio of the
measured OS signals in mixed extracts to the sum of OS signals measured in each extract before mixing,
are used to evaluate the matrix effect of OSs. A $F_{matrix}$ value of less than 1 indicates the presence of
matrix effect.
As shown in Figure S2, the $F_{matrix}$ values were significantly smaller than 1 in both summer and
winter, indicating that the signal responses of 2-MT-OS and 2-MG-OS in mixed extracts were largely
suppressed due to the matrix effect. Notably, $F_{matrix}$ exhibits a significant negative dependence on the
reduced mass ($\mu$, $\mu g\ m^{-3}$), a proxy used to represent effective mass loadings of the mixed $PM_{2.5}$ extracts,
defined as:
$$\mu = \sqrt{m_1 * m_2/(m_1 + m_2)} \tag{1}$$
where $m_1$ and $m_2$ are the $PM_{2.5}$ mass loading of individual samples. This observation suggests that the
concentrations of iSOA OSs in $PM_{2.5}$ samples collected in 2015 were underestimated more than those
in 2021, given that ambient $PM_{2.5}$ concentrations declined by 39.8% and 47.0% from 2015 to 2021 in
summer and winter, respectively.
As $PM_{2.5}$ concentrations in ambient samples used for the matrix effect evaluation generally
represent lower or upper ends of $PM_{2.5}$ concentrations during the observation period (see Table S3), the
relative differences in measured $F_{matrix}$ values at varying reduced mass (Figure S2) may roughly reflect
the differences in the extent of underestimation in OS concentrations for samples collected across 2015-
2021. During summer, the $F_{matrix}$ value decreased from 0.71 to 0.63 for 2-MT-OS and from 0.85 to 0.58
for 2-MG-OS with increasing reduced mass, indicating that the concentrations of these two iSOA OSs
were a factor of 1.2 and 1.5 more underestimated in 2015 than in 2021 due to matrix effect. Similarly,
during winter the $F_{matrix}$ values of iSOA OSs decreased from 0.9 to 0.6 with rising reduced mass,
implying a factor of 1.5 greater underestimation in OS concentrations in 2015 than in 2021.
**2.4 Estimation of the reaction efficiency of isoprene-derived epoxides in aqueous aerosols**
The IEPOX/HMML/MAE can undergo acid-catalyzed nucleophilic addition with water and sulfate
to form polyol tracers (2-MTs and 2-MG) and OSs (2-MT-OS and 2-MG-OS) in aqueous aerosols,
respectively. The aqueous-phase pseudo-first-order rate constants ($k_{aq}$, $s^{-1}$) for epoxides could be
estimated by eq 1 (Eddingsaas et al., 2010),
$$k_{aq} = \sum_{i=1}^{N} \sum_{j=1}^{M} k_{ij}[nuc_i]_{aq}[acid_j]_{aq} \tag{2}$$
Where $k_{ij}$ is the third-order reaction rate constant of isoprene-derived epoxides with nucleophile i
(water or sulfate) and acid j (hydrogen ion or bisulfate) in the aqueous phase and the reported values of
$k_{ij}$ in previous studies are shown in Table 1. When epoxides react with water and sulfate, the aqueous-
phase reaction rate constants ($k_{aq,H_2O}$ and $k_{aq,SO_4^{2-}}$) can be estimated by eqs 2 and 3, respectively.
$$k_{aq,\ H_2O} = k_{H_2O,H^+}[H_2O]_{aq}[H^+]_{aq} + k_{H_2O,HSO_4^-}[H_2O]_{aq}[HSO_4^-]_{aq} \quad (3)$$
$$k_{aq,\ SO_4^{2-}} = k_{SO_4^{2-},H^+}[SO_4^{2-}]_{aq}[H^+]_{aq} + k_{SO_4^{2-},HSO_4^-}[SO_4^{2-}]_{aq}[HSO_4^-]_{aq} \quad (4)$$
Where $[H^+]_{aq}$, $[HSO_4^-]_{aq}$, and $[H_2O]_{aq}$ are molar concentrations of hydrogen ion, bisulfate, and
ALWC in aqueous aerosols, which were estimated by ISORROPIA-II model, and $[SO_4^{2-}]_{aq}$ is the molar
concentration of sulfate.

The reactive uptake coefficient (γ) of epoxides on aqueous aerosols is parameterized by a resistor

model (Xu et al., 2016; Pye et al., 2013).
$$\frac{1}{\gamma} = \frac{1}{\alpha} + \frac{\omega}{4H_{epoxide}RT\sqrt{D_a k_{aq}}} \frac{1}{f(q)} \quad (5)$$
$$f(q) = \coth(q) - \frac{1}{q} \quad (6)$$
$$q = r_p \sqrt{k_{aq}/D_a} \quad (7)$$
Where $r_p$ is the particle radius. Previous work has found that the surface area ($S_a$, m$^2$ m$^{-3}$) and volume
concentrations ($V_a$, m$^3$ m$^{-3}$) of dry PM$_{2.5}$ could be described as a function of PM$_{2.5}$ mass concentration
($C_{PM_{2.5}}$, μg m$^{-3}$) in Shanghai ($S_a = 7.54 \times 10^{-6} \cdot C_{PM_{2.5}} + 1.01 \times 10^{-4}$, $V_a = 5.59 \times 10^{-13} \cdot C_{PM_{2.5}} + 1.02 \times 10^{-12}$)
(Zang et al., 2022). In this study, the mean particle radius of dry PM$_{2.5}$ was calculated as $3V_a/S_a$, which
was then corrected for the aerosol hygroscopic growth to get the wet particle radius based on the *κ*-
Köhler hygroscopicity function (see details in Section S4). α is the mass accommodation coefficient
taking a value of 0.02 for both IEPOX and HMML&MAE (Mcneill et al., 2012), ω is the mean
molecular velocity of epoxides, H$_{epoxide}$ is the Henry's law constant in the aqueous phase, with a value
of $2.7 \times 10^6$ M atm$^{-1}$ for IEPOX (Pye et al., 2013) and a constrained value of $7.5 \times 10^6$ M atm$^{-1}$ for
HMML&MAE by CMAQ in Case 1, and k$_{aq}$ is the first-order reaction rate constant in the aqueous
phase (s$^{-1}$), estimated using eq 1.
To describe the overall loss rate of gas-phase epoxides due to the reactive uptake by aqueous
aerosols, the pseudo-first-order heterogeneous reaction rate constant was calculated by eq 7, when
neglecting the gas-phase diffusion limitation:
$$k_{het} = \gamma \omega S_a / 4 \qquad (8)$$

**2.5 Model Simulations**
The CMAQ model (v5.2) was adopted to simulate the gas-phase concentration of isoprene and
particulate concentrations of 2-MTs, 2-MG, and their OS derivatives formed involving the reactive
uptake of IEPOX and HMML&MAE on aqueous aerosols (Pye et al., 2017) in both summer and winter
of 2015 and 2019 in Shanghai. The simulations performed with the standard CMAQ v5.2 are referred
to as the Base Case. While the advanced model simulations performed according to a recent study by
Zhang et al. (2023) are named as Case 1. In this advanced case, the iSOA polyol tracers were treated
as semi-volatile species that partition between gas, aqueous, and organic phases, while the OS tracers
were treated as non-volatile species. The removal of iSOA polyol tracers by OH radicals in the gas and
particle phases was also considered. The key parameters for simulating reactive uptake of
IEPOX/HMML&MAE and the removal of 2-MTs and 2-MG in the gas phase and aqueous aerosol in
the model are listed in Table S4 and S5.
In this work, three nested domains covering mainland China, eastern China, and the Yangtze
River Delta were configured with horizontal resolutions of 36 km × 36 km (d01), 12 km × 12 km (d02),
and 4 km × 4 km (d03), as shown in Figure S3. The outermost domain (d01) was driven by predefined
initial and boundary conditions from CMAQ, with its outputs supplying these conditions for d02,
which in turn provided them for d03. All domains employed a vertical structure consisting of 18 layers,
extending from the surface to an altitude of 21 km.
Anthropogenic emissions were sourced from the Multi-resolution Emission Inventory for China
(MEIC) version 1.4 for China (Geng et al., 2024) and the Regional Emission inventory in ASia (REAS)
version 3.2.1 for other Asian countries and regions (Kurokawa and Ohara, 2020). Open biomass
burning emissions were based on the Fire INventory from NCAR (FINN) version 2.5 (Wiedinmyer et
al., 2023). Biogenic emissions were estimated using the Model of Emissions of Gases and Aerosols
from Nature (MEGAN) version 2.1, incorporating the high-quality Leaf Area Index (HiQ-LAI) dataset
developed by Yan et al. (2024), which enhances the spatiotemporal consistency of MODIS LAI
products. Meteorology data was generated by the Weather Research and Forecasting (WRF) model
version 4.2.1 with initial and boundary conditions from the fifth generation ECMWF atmospheric
reanalysis data (ERA5).
The simulation was conducted for four periods: 12 July–9 August 2015, 25 December 2015–16
January 2016, 4–26 January 2019, and 12 July–5 August 2019. The first two days of each period were
treated as spin-up periods and excluded from the analysis.
**3. Results and Discussion**
**3.1 Temporal evolution of major air pollutants during the observation period**
Time series of $PM_{2.5}$ and its major components, as well as the major trace gases ($NO_2$ and $O_3$) and
meteorological parameters during the observation period are shown in Figure 1 and the seasonal and
inter-annual variations of various pollutants are further demonstrated in Figure 2. The concentration of
$NO_2$ exhibited an obvious downward trend, in particular in summer from 2015 to 2021, consistent with
a strong reduction in anthropogenic emissions during this period. By contrast, the concentration of $O_3$
significantly decreased from $52.0 \pm 38.9$ ppb in 2015 to $41.2 \pm 22.8$ ppb in 2019 ($p < 0.05$) and then
remained at a comparable level ($43.4 \pm 20.8$ ppb) in 2021, suggesting a complex response of secondary
$O_3$ formation to primary emission reductions. During the observation period, the average $PM_{2.5}$
concentration decreased by 41.7% from 2015 to 2021, with concentrations of major components,
including sulfate, ammonium, and OM, decreasing by 51.8%, 40.6%, and 39.1%, respectively (Figure
2). In contrast, the concentration of nitrate showed a slight upward trend during this period, in line with
the measurement in urban Shanghai by Zhou et al. (2022). Overall, OM was the most abundant
component in $PM_{2.5}$, accounting for 10.2-72.7% (average 22.6%) of total $PM_{2.5}$ mass, followed by
sulfate (6.8-45.2%, average 17.7%), nitrate (0.5-32.6%, average 17.0%), and ammonium (1.1-18.2%,
average 8.8%). Ascribed to the strong decrease of inorganic ion concentrations (in particular sulfate),
aerosol LWC decreased from $9.14 \pm 4.51$ $\mu g\ m^{-3}$ in 2015 to $4.40 \pm 2.76$ $\mu g\ m^{-3}$ in 2021 ($p < 0.05$).
Aerosol pH decreased from $3.2 \pm 0.4$ in 2015 to $2.5 \pm 0.9$ in 2021 ($p < 0.05$), which was mainly driven
by the decrease of non-volatile cations during these years, though the decreased concentrations of
sulfate had an opposite effect on aerosol acidity (Zhou et al., 2022).

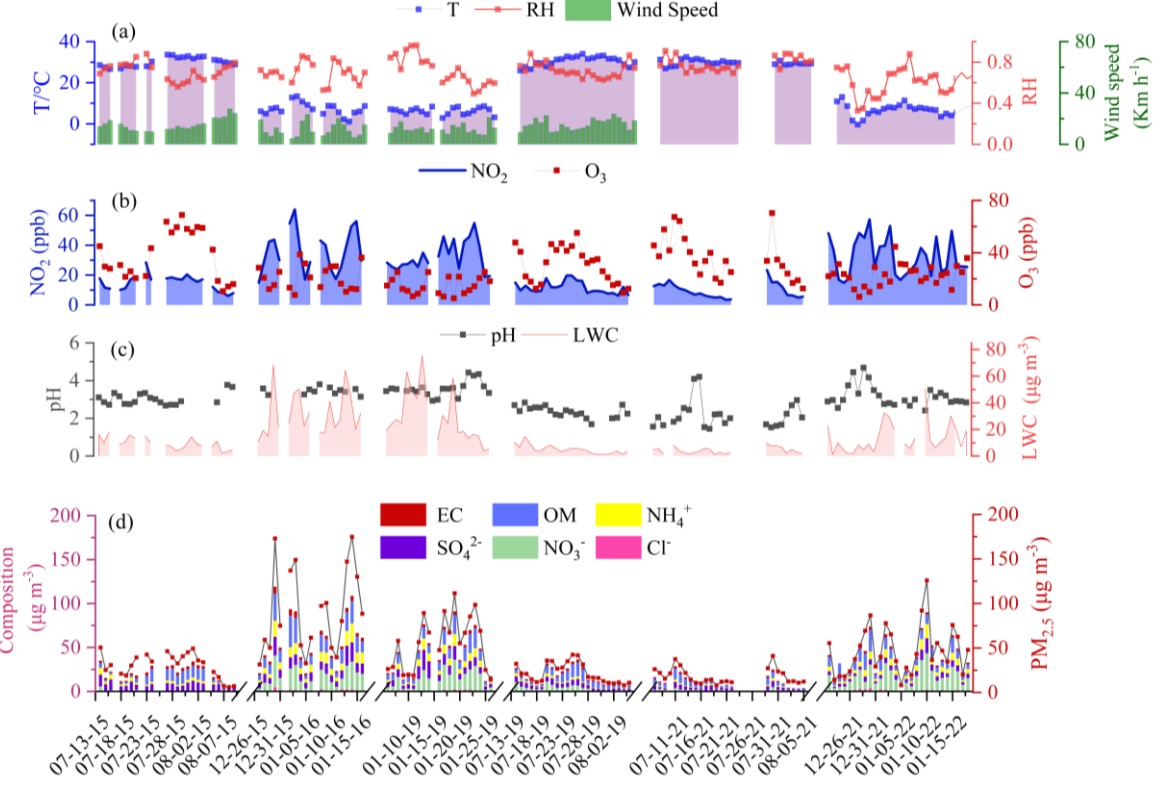


Figure 1. Temporal variations of (a) meteorological parameters (ambient temperature, relative humidity, and wind speed), (b) concentrations of $NO_2$ and $O_3$, (c) aerosol pH and liquid water content (LWC), and (d) concentrations of $PM_{2.5}$ and its major components (OM, EC, sulfate, nitrate, chloride, and ammonium) in urban Shanghai during the observation period. The wind speed data were not collected during the observations in 2021.

## 3.2 Seasonal and annual variations of iSOA tracers

The measured concentrations of particulate iSOA tracers during the observation period are summarized in Figure 2 and their specific concentration values are also provided in Table S1. Among the measured iSOA tracers, $C_5$-alkene triols were the most abundant species with average concentrations of 27.6 ng m$^{-3}$ in 2015, 20.9 ng m$^{-3}$ in 2019, and 11.1 ng m$^{-3}$ in 2021, accounting for 28.8 %, 22.4, and 18.7 % of the total iSOA mass, followed by 2-MT-OS, 2-MTs, 2-MG-OS, and 2-MG. However, the concentrations of $C_5$-alkene triols might be overestimated since previous studies have reported that concentrations of $C_5$-alkene triols could be artifacts of thermal degradation products of 3-methyltetrahydrofuran-2,4-diols and 2-MT-OS during GC/MS analysis (Cui et al., 2018; Frauenheim et al., 2022). Frauenheim et al. (2022) found that less than 15% of 3-methyltetrahydrofuran-2,4-diols could

transfer to two isomers of $C_5$-alkene triols (cis-/trans-3-methyl-but-3-ene-1,2,4-triols). In the present
study, using 2-methylerythritol as a surrogate standard, the concentrations of 3-methyltetrahydrofuran-
2,4-diols were determined to be less than 5% of $C_5$-alkene triols in summer but had comparable
concentrations to $C_5$-alkene triols in winter. This result indicates that 3-methyltetrahydrofuran-2,4-diols
was a minor contributor to $C_5$-alkene triols in summer but an important source for $C_5$-alkene triols in
winter. In contrast, the contribution from the thermal degradation of 2-MT-OS might be more significant,
though the specific contribution remains to be quantified; Cui et al. (2018) found that thermal
degradation of 2-MT-OS could generate all three isomers of $C_5$-alkene triols, while (Yee et al., 2020)
found that the thermal decomposition of 2-MT-OS could only produce one isomer, 3-methyl-2,3,4-
trihydroxy-1-butene. Given these uncertainties, it is difficult to quantitatively evaluate the artifact
formation of $C_5$-alkene triols during GC/MS analysis. Therefore, the abundance and inter-annual trend
of $C_5$-alkene triols were not discussed in detail in the present work.

As shown in Figure 2, the particle-phase concentrations of total and specific IEPOX-SOA tracers

(except 2-MTs) decreased yearly in both summer and winter between 2015-2021 (Figs. 2a-d), while the
particulate concentrations of total and specific HMML&MAE-SOA species did not show a significant
inter-annual trend in summer during this period (Figs. 2e-g). However, the inter-annual trend of iSOA
OSs could be altered due to the matrix effect. The measured concentration of 2-MT-OS exhibited a
decreasing inter-annual trend, while 2-MG-OS showed insignificant variation between 2015-2021.
Accounting for the significantly larger matrix effects in 2015 samples compared to 2021 samples (see
Section 2.3), the true concentrations of 2-MT-OS would decrease more sharply and 2-MG-OS might
also exhibit a declining trend. Recently, Frauenheim et al. (2024) demonstrated that gas-phase OH
oxidation of 3-methylenebutane-1,2,4-triol (3-MBT) and 3-methyltetrahydrofuran-2,4-diols, formed
from acid-catalyzed isomerization of IEPOX in aerosols, can yield 2-MT-OS and 2-MG-OS. However,
in the present study, the gas-phase concentrations of 3-MBT and 3-methyltetrahydrofuran-2,4-diols
were not measured, precluding a quantitative assessment of the contribution of their oxidation products
to 2-MT-OS and 2-MG-OS. Given that IEPOX-SOA concentrations observed here exhibited a
significant decreasing trend over the observation period, the concentrations of IEPOX isomerization
products, such as 3-MBT and 3-methyltetrahydrofuran-2,4-diols, likely also decreased annually.
Therefore, gas-phase OH oxidation of these species might represent a plausible contributor to the inter-
annual decline in 2-MT-OS and 2-MG-OS.

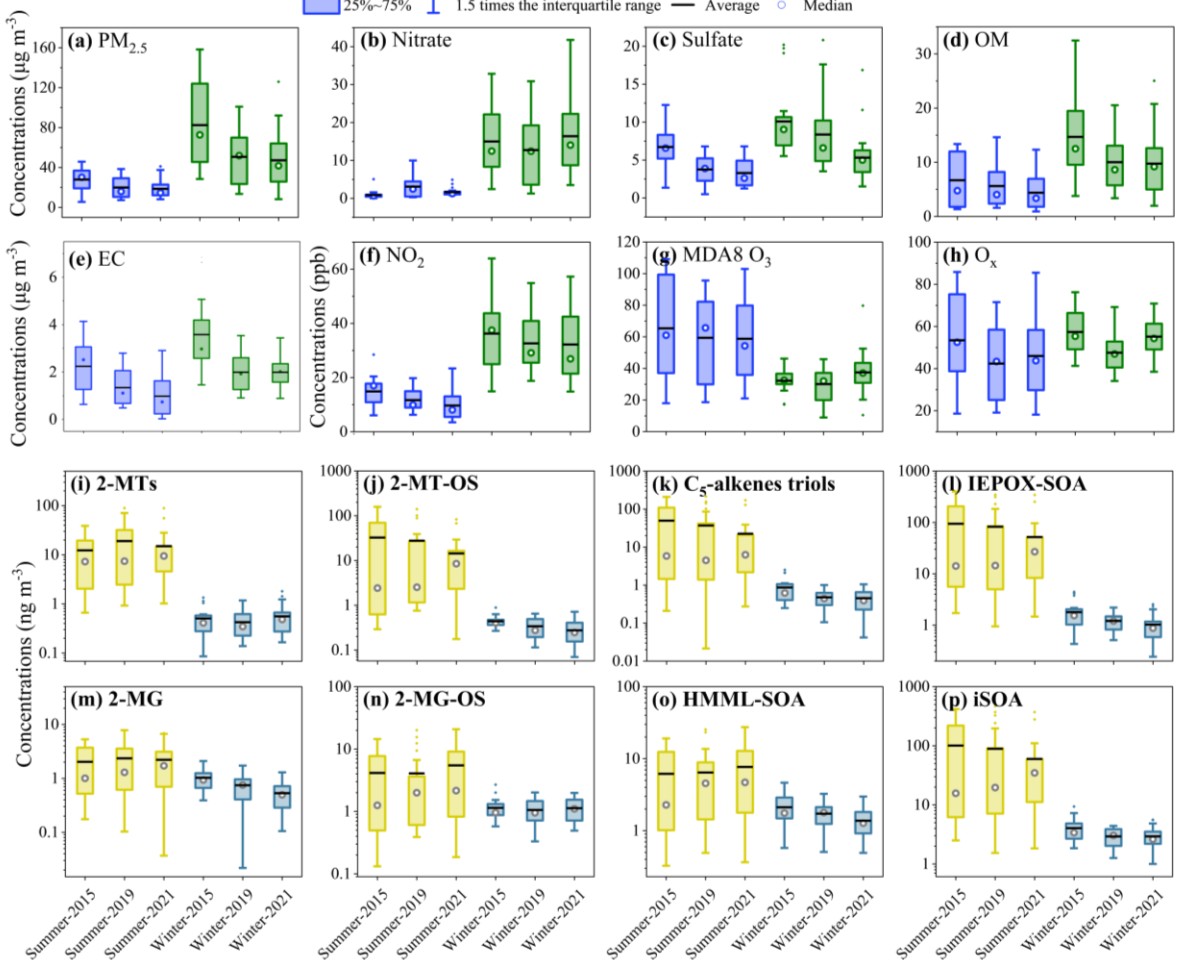


Figure 2. Seasonal and inter-annual variations concentration of PM$_{2.5}$ and its major components (a-e),
gas-phase anthropogenic pollutants (f-h), as well as particulate iSOA tracers, including (i)2-MTs, (j) 2-
MT-OS, (k) C$_5$-alkene triols, (l) IEPOX-SOA (the sum of 2-MTs, 2-MT-OS, and C$_5$-alkene triols), (m)
2-MG, (n) 2-MG-OS, (o) HMML&MAE-SOA (2-MG plus 2-MG-OS), and (p) iSOA (the sum of all
tracers).
The particulate concentrations of IEPOX-SOA (52.6 ± 97.6, 46.7 ±103.8, and 25.8 ± 62.9 ng m$^{-3}$
in 2015, 2019 and 2021, respectively) dominated over HMML&MAE-SOA (6.2 ± 6.4, 6.4 ± 7.0, and
7.7 ± 7.7 ng m$^{-3}$) in summer, while they were comparable to HMML&MAE-SOA in winter. Although
C$_5$-alkene triols might be largely artifacts of GC/MS analysis (Cui et al., 2018; Frauenheim et al., 2022),
the concentrations of IEPOX-SOA excluding C$_5$-alkene triols were still dominant over HMML&MAE-
SOA in summer. In addition, previous studies have demonstrated that the concentrations of 2-MT-OS
were underestimated more than 2-MG-OS by a factor of 5.7-9.1 in Beijing (Bryant et al., 2020) and 2.9
in Guangzhou (Bryant et al., 2021). If 2-MT-OS was also more significantly underestimated than 2-
MG-OS in the present study, the predominance of IEPOX-SOA over HMML&MAE-SOA would be
more pronounced. The dominance of IEPOX-SOA over HMML&MAE-SOA in summer is in
agreement with RPLC-MS measurements in Beijing, Hefei and Kunming in China (Zhang et al., 2022)
and Birmingham, US (Rattanavaraha et al., 2016), as well as hydrophilic interaction liquid
chromatography-mass spectrometry (HILIC-MS) measurements conducted in urban Guangzhou, China
(Liu et al., 2025).

The annual average particulate concentration of the total iSOA polyol tracers (including 2-MTs, 2-

MG, and $C_5$-alkene triols) were $36.1 \pm 63.3$, $33.4 \pm 70.5$, and $18.7 \pm 47.1$ ng m$^{-3}$ in 2015, 2019 and 2021,
higher than those of OS tracers (including 2-MT-OS and 2-MG-OS) by a factor of 2.5, 2.1, and 1.4,
respectively. However, given the significant underestimation of iSOA OSs due to matrix effect and
overestimation of $C_5$-alkene triols due to their potential artifact formation, the true concentration of
iSOA OSs would predominate over that of polyol tracers. The iSOA OSs prevailing over polyol tracers
is consistent with urban observations using HILIC-MS, such as in Manaus, Brazil (Cui et al., 2018) and
Guangzhou, China (Liu et al., 2025) (see Table S6). Using RPLC-MS, Bryant et al. (2020) also observed
higher concentrations of iSOA OSs than polyol tracers in Beijing, China. Considering the potential
underestimation of iSOA OSs due to matrix effect, the concentration of iSOA OSs would be even higher
than that of polyol tracers in their study.

2-MTs exhibited a different inter-annual trend compared to other IEPOX-SOA tracers (Figure 2i).

A possible explanation for this discrepancy is that 2-MTs may have origins other than reactive uptake
of IEPOX on aqueous aerosol. Previous studies have found that 2-methylerythritol, one isomer of 2-
MTs, could be generated by biosynthetic pathways (Duvold et al., 1997; Sagner et al., 1998; Rohmer,
1999; Lange et al., 2000; Yang et al., 2013). And the contributions of non-IEPOX pathway to 2-MTs
concentrations were pH-dependent, accounting for 20-40% in areas with aerosol pH < 2 and more than
70% under less acidic conditions (pH ~ 2–5) (Zhang et al., 2023). The contribution of biological
emissions to 2-MTs might be important in Shanghai given its less acidic aerosol conditions (pH > 2).
As a whole, the particulate concentrations of iSOA decreased significantly in both seasons from 2015
to 2021. In addition, all iSOA compound classes had substantially higher concentrations in summer than
in winter. Such a strong seasonality in abundance is mainly driven by the higher temperature and
stronger solar radiation, and thereby more intensive isoprene emissions and photochemistry in summer
than in winter. Notably, HMML&MAE-SOA species exhibited a relatively smaller seasonal variation
than IEPOX-SOA. This is partially owing to the fast thermolysis of methacryloyl peroxynitrate (MPAN)
in summer, reducing the formation of HMML&MAE and thereby SOA (Worton et al., 2013).
To further investigate factors that affect the abundance of iSOA, the particulate concentrations of
2-MTs, 2-MT-OS, 2-MG, and 2-MG-OS, as well as the gas-phase concentration of isoprene were
simulated with the CMAQ model and compared to measurements in summertime and wintertime of
2015 and 2019 (see Figure 3). Predicted concentrations of isoprene were generally consistent with
observations with a median correlation ($r^2$=0.45), except in the summer of 2015, during which isoprene
concentrations were significantly overestimated (Figure 3a). For iSOA tracers, the Case 1 showed a
better prediction than the Base Case. Overall, the simulated IEPOX-SOA tracers were biased low in
summer, but biased high in winter (Figure 3b and 3c). In contrast, the 2-MG and 2-MG-OS were biased
low in both seasons (Figure 3d and 3e). The underestimation of 2-MG is consistent with previous
simulations at 14 sites across China in the summer of 2012 (Qin et al., 2018). Accounting for the
underestimation of OSs due to the matrix effect, simulated concentrations of 2-MT-OS would be more
biased low in summer but might be close to observations in winter. Similarly, the under-prediction of
2-MG-OS would be more significant in both seasons. The larger uncertainty of simulated
HMML&MAE-SOA tracers compared to that of IEPOX-SOA might be attributed to the lack of well
constrained kinetic parameters for reactive uptake of HMML and MAE. In addition, simulated
concentrations of iSOA tracer species had decreasing inter-annual trend in both Base Case and Case 1
(Figure S4), which was in agreement with observations of total iSOA, 2-MT-OS, and 2-MG-OS (with
matrix effect considered), but not for 2-MG and 2-MTs. These results suggest that the major factors
driving the overall formation and evolution of iSOA and in particular iSOA OSs were captured by the
model, while some factors governing the abundance of 2-MG and 2-MTs might not be well represented
in the model.

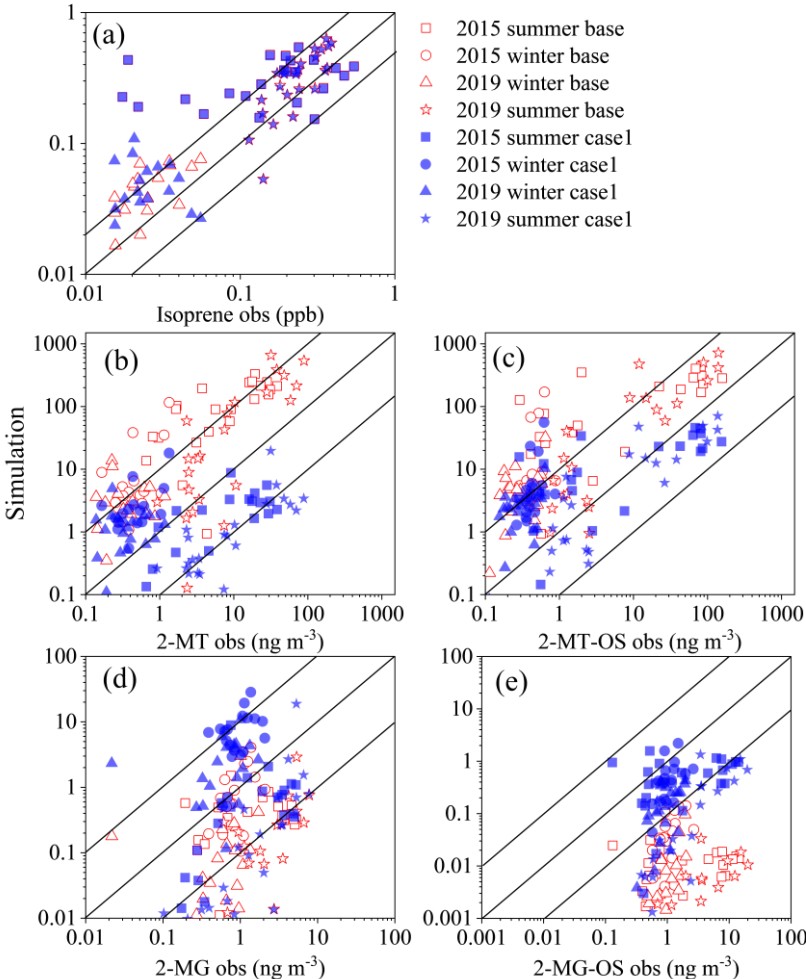

Figure 3. Comparisons of simulations against observations for (a) isoprene, (b) 2-MTs, (c) 2-MT-OS, (d) 2-MG, (e) 2-MG-OS in summer and winter of 2015 and 2019. Red dot represents simulations with standard CMAQ v5.2 (Base Case), and blue represents simulations using the optimized model (Case 1). The 1:1, 10:1, and 1:10 lines are shown with solid lines.

The gas-phase and particle-phase fractions of 2-MG and 2-MTs were estimated using a chemical equilibrium partitioning model as described in Section S3 and the results are shown in Figure S5. The particle-phase fraction ($F_p$) of 2-MG was highly variable, with a significant lower value in summer (9.0 -19.0%) than in winter (31.6 - 44.0%), indicating substantial amounts of 2-MG was present in the gas phase in summer. Additionally, the $F_p$ value of iSOA polyols, in particular for 2-MG, decreased yearly. As a result, the gas-plus-particle-phase concentrations of 2-MG showed an upward trend in summer from 2015-2021 (p < 0.05, Figure S5). In contrast, 2-MTs were mainly distributed in the particle phase with $F_p$ values larger than 70% in both seasons, consistent with previous measurements (Isaacman-

Vanwertz et al., 2016). The inter-annual trend of 2-MTs was not significantly affected by their gas-phase
fraction because of the relatively low volatility. Overall, compared to other iSOA tracers in summer, 2-
MG and 2-MTs exhibited a distinctly different inter-annual trend over 2015-2021. It should be noted
that the use of surrogate standards would not alter the trend of iSOA tracers. The key factors driving
such trends will be discussed in detail below.
**3.3 Key influencing factors of iSOA formation**
The production of iSOA can be influenced by a variety of factors such as the emission and
concentration of isoprene, atmospheric oxidizing capacity as represented by the concentrations of $O_3$ or
odd oxygen ($O_x = O_3 + NO_2$), nitrogen oxides, as well as aerosol composition and properties including
sulfate content, acidity, and LWC. Here, we identify the major influencing factors of IEPOX-SOA and
HMML&MAE-SOA formation through the correlation analysis between different iSOA tracers and
influencing factors (Figure 4).
The correlations of all iSOA species with isoprene were relatively weak, with most of the
correlation coefficients ($r^2$) below 0.37. Therefore, the decline in iSOA concentrations from 2015 to
2021 could not be attributed to the slight variation in isoprene concentration. The HMML&MAE-SOA
species exhibited strong correlations with ozone ($r^2 = 0.48$-$0.81$) and $O_x$ ($r^2 = 0.57$-$0.82$) in summer, in
particular in 2015 and 2021 while exhibiting relatively weaker correlations with $NO_2$ ($r^2 = 0.20$-$0.55$).
Such correlations between 2-MG and ozone were also observed in previous measurements in urban
areas in southeastern US (Rattanavaraha et al., 2016), which proposed that ozone might be a superior
indicator to $NO_x$ for the photochemical process of isoprene under $NO_x$-dominant conditions. The
IEPOX-SOA tracers also correlated well with ozone ($r^2 = 0.36$-$0.70$) or $O_x$ ($r^2 = 0.40$-$0.68$) in summer,
despite less strongly than HMML&MAE-SOA species. These observations clearly suggest that
atmospheric oxidation capacity (or the oxidation of isoprene to epoxide intermediates) plays a driving
role in summertime iSOA formation. In addition, weak to moderate correlations ($r^2 = 0.23$-$0.45$) were
observed between the iSOA tracers and sulfate aerosol in 2019 and 2021, indicating that sulfate aerosol
also plays a role in controlling iSOA formation during these periods. In contrast, wintertime iSOA
species exhibited weak correlations with $O_3$ and $O_x$ in all the three years (Figure 4d-f). However, they
all had moderate or strong correlations with sulfate aerosol ($r^2 = 0.36$-$0.68$) in 2015 and the iSOA OSs

correlated moderately with sulfate ($r^2 = 0.35\text{-}0.58$) and LWC ($r^2 = 0.34\text{-}0.58$) in 2019 and 2021. These results suggest that sulfate-mediated heterogeneous chemistry of isoprene epoxide intermediates in aqueous aerosols is a key process controlling iSOA formation in winter (Surratt et al., 2010; Yee et al., 2020; Lin et al., 2012). A sensitivity test considering the measurement uncertainties of iSOA tracers did not significantly influence the correlation analysis results (see details in Section S5).

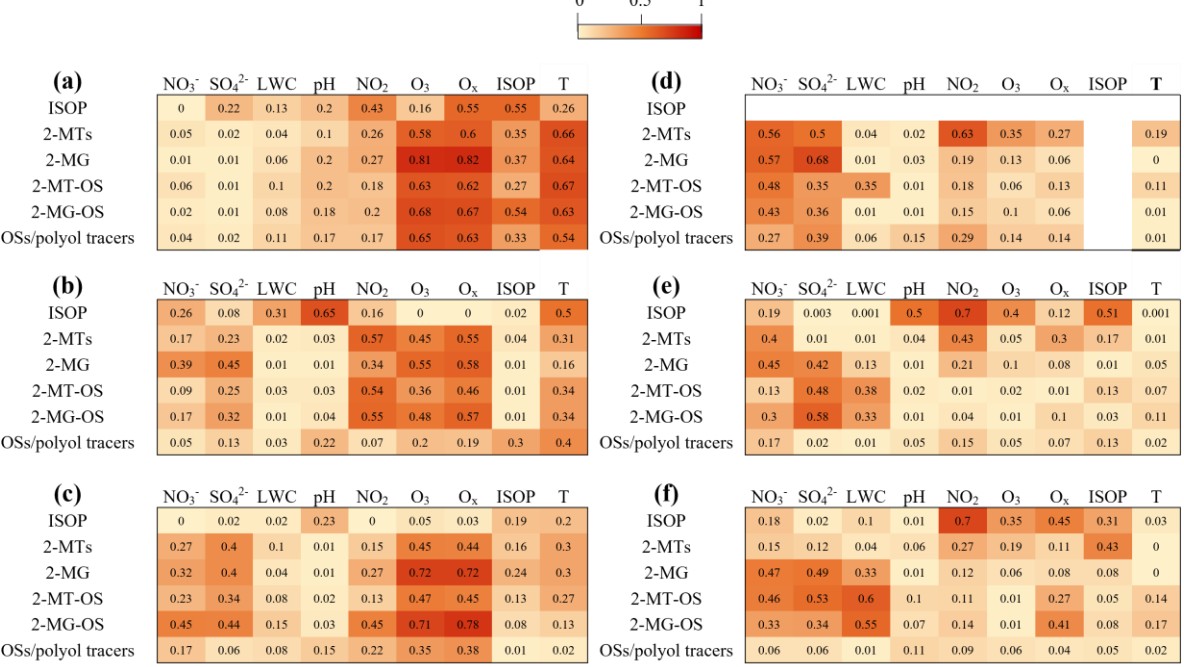

Figure 4. Coefficients of correlation ($r^2$) between iSOA compounds and various influencing factors of iSOA formation in (a-c) summer and (d-f) winter of 2015, 2019, and 2021, respectively. The ISOP is the abbreviation of isoprene.

As shown in Table S1, the concentrations of sulfate aerosol and LWC both decreased drastically over 2015-2021 ($p < 0.05$), which could explain the declining trend of iSOA in winter and partially contributed to the decreased formation of IEPOX-SOA in summer. Notably, the summertime average MDA8 (maximum daily 8-h average) $O_3$ concentration at the observation site increased from 2015 to 2017 and then decreased significantly in 2018, followed by a slight increasing trend between 2018 and 2021 (Figure S6). This inter-annual trend of $O_3$ was similar to the trend of annual 90[th] percentile MDA8 $O_3$ concentration in the region of Shanghai (Figure S6). As a result, the summertime average MDA8 $O_3$ concentration in 2015, 2019, and 2021 showed a slight downward trend. In addition, the simulated concentrations of OH radicals and ratios of (MVK+MACR)/isoprene both declined in 2019 compared to those in 2015 (see Table S8), indicating a decline in the atmospheric oxidation capacity during the

observation period. Similarly, the nighttime atmospheric oxidation capacity as indicated by the production rate of $NO_3$ radicals (PNO$_3$), calculated by multiplying the reaction rate coefficient between $NO_2$ and $O_3$ by their concentrations (Wang et al., 2023a), also decreased during this period (Figure S7). The reduced atmospheric oxidation capacity further explained the decreased formation of iSOA OSs in summer during these years, but it could not explain the inter-annual variation in summertime 2-MG and 2-MTs, suggesting that other factors might have offset the anticipated decline in these polyols during this period.

**3.4 Heterogeneous reactivity of ambient aerosols**

To better understand the role of heterogeneous chemistry in the formation and inter-annual variations of iSOA, the reactive uptake coefficients ($\gamma_{EPOXIDE}$) of isoprene-derived epoxides on ambient aerosols were estimated by a resistor model (eq. 4) (Xu et al., 2016; Pye et al., 2013). The pseudo-first-order heterogeneous reaction rate constant ($k_{het}$, s$^{-1}$) of gas-phase IEPOX and HMML&MAE could be then estimated from $\gamma_{EPOXIDE}$ via eq. 7. Currently, there are five sets of reported third-order reaction rate constants (i.e., $k_{i,j}$ in eq. 1) for the acid-catalyzed nucleophilic addition of water and sulfate to IEPOX in the aqueous phase (Table 1). Piletic et al. (2013) predicted $k_{i,j}$ for IEPOX ($k_{i,j\text{-IEPOX}}^{-1}$) with a computational model, which are two orders of magnitudes higher than the laboratory-measured values ($k_{i,j\text{-IEPOX}}^{-2}$) by Riedel et al. (2015) and model-estimated values ($k_{i,j\text{-IEPOX}}^{-3}$) using CMAQ by Pye et al. (2013). More recently, Pye et al. (2017) updated the values ($k_{i,j\text{-IEPOX}}^{-4}$) by constraining the $k_{i,j\text{-IEPOX}}^{-3}$ using measured 2-MT-OS/2-MTs in CMAQ and Chen et al. (2024) constrained reaction rate constant of IEPOX ($k_{i,j\text{-IEPOX}}^{-5}$) using a phase-separation box model with chamber measurements. For HMML and MAE, there is a lack of direct measurements and theoretical calculations of their $k_{i,j}$ values. Pye et al. (2013) assumed the same $k_{i,j}$ values as IEPOX with $k_{i,H+}$ of $9.0 \times 10^{-4}$ M$^{-2}$ s$^{-1}$ for water and $2.0 \times 10^{-4}$ M$^{-2}$ s$^{-1}$ for sulfate.

Table 1. Third-order reaction rate constants of IEPOX and HMML&MAE with sulfate and water in
the aqueous phase

| | $k_{i,H+}$ (M$^{-2}$·s$^{-1}$) | | $k_{i,HSO4^-}$ (M$^{-2}$·s$^{-1}$) | | References |
|---|---|---|---|---|---|
| | i=H$_2$O | i=SO$_4^{2-}$ | i=H$_2$O | i=SO$_4^{2-}$ | |
| $k_{i,j\text{-IEPOX}}^{-1}$ | $5.3\times10^{-2}$ | $5.2\times10^{-1}$ | — | — | (Piletic et al., 2013) |
| $k_{i,j\text{-IEPOX}}^{-2}$ | $3.4\times10^{-4}$ | $4.8\times10^{-4}$ | — | — | (Riedel et al., 2015) |
| $k_{i,j\text{-IEPOX}}^{-3}$ | $9\times10^{-4}$ | $2\times10^{-4}$ | $1.3\times10^{-5}$ | $2.9\times10^{-6}$ | (Pye et al., 2013) |
| $k_{i,j\text{-IEPOX}}^{-4}$ | $9\times10^{-4}$ | $8.8\times10^{-3}$ | $1.3\times10^{-5}$ | $2.9\times10^{-6}$ | (Pye et al., 2017) |
| $k_{i,j\text{-IEPOX}}^{-5}$ | $5.3\times10^{-4}$ | $5.2\times10^{-3}$ | — | — | (Chen et al., 2024) |
| $k_{i,j\text{-HMML\&MAE}}$ | $9\times10^{-4}$ | $2\times10^{-4}$ | $1.3\times10^{-5}$ | $2.9\times10^{-6}$ | (Pye et al., 2013) |

Firstly, the $k_{i,j}$ values of IEPOX and HMML&MAE were evaluated by comparing the measured
ratios of 2-MT-OS/2-MTs and 2-MG-OS/2-MG with the calculated ratios of the pseudo-first-order rate
constants for the nucleophilic addition reactions of epoxides with sulfate and water ($k_{aq, SO_4^{2-}}/k_{aq, H_2O}$).
The results are shown in Figure S8 and S9. For IEPOX, the $k_{aq, SO_4^{2-}}/k_{aq, H_2O}$ ratios were close to
measured particulate ratios of 2-MT-OS/2-MTs when using $k_{i,j\text{-IEPOX}}^{-1}$, $k_{i,j\text{-IEPOX}}^{-4}$, and $k_{i,j\text{-IEPOX}}^{-5}$ suggested
by Piletic et al. (2013), Pye et al. (2017), and Chen et al. (2024), respectively (Figure S8). However,
when taking into account the underestimation of 2-MT-OS due to matrix effect, all estimated ratios were
lower than the measured values. Similarly, the calculated $k_{aq, SO_4^{2-}}/k_{aq, H_2O}$ ratios for HMML and MAE
were 1-2 orders of magnitude lower than the measured 2-MG-OS/2-MG ratios (Figure S9) and such a
discrepancy would be larger when considering the matrix effect of 2-MG-OS. This result indicates that
the $k_{ij}$ of isoprene-derived epoxides, particularly that of HMML and MAE, with sulfate was likely
underestimated, since Pye et al. (2013) found their hydrolysis rate constant allowed a good prediction
of the concentration of 2-MG.
Figure 5 shows the γ and $k_{het}$ values for IEPOX and HMML&MAE estimated using different sets
of kinetic parameters listed in Table 1. The CMAQ-modeled values are also displayed for comparison.
The $k_{het}$ estimated by $k_{i,j\text{-IEPOX}}^{-1}$, $k_{i,j\text{-IEPOX}}^{-2}$ and $k_{i,j\text{-IEPOX}}^{-5}$ in summer had highest values in 2019. This
might be attributed to the fact that these three sets of parameters lack the third-order reaction rate
constant of IEPOX with nucleophiles catalyzed by bisulfate (Table 1). As a comparison, we calculated
the $k_{het}$ of IEPOX and HMML&MAE excluding the reaction rate constant catalyzed by bisulfate using
$k_{i,j-IEPOX}^{-3}$ and $k_{i,j-IEPOX}^{-4}$ (Figure S10). We found that the inter-annual trend of IEPOX (Figure S10a) and
HMML&MAE (Figure S10b) in summer was altered and similar to that of IEPOX calculated by $k_{i,j-}$
$_{IEPOX}^{-1}$, $k_{i,j-IEPOX}^{-2}$ and $k_{i,j-IEPOX}^{-5}$, while the inter-annual trend of $k_{het}$ in winter was not sensitive to the
exclusion of reaction rate constant catalyzed by bisulfate. This result indicates a contribution of
nucleophilic-addition of epoxides catalyzed by bisulfate to the heterogeneous reactivity. It also suggests
that the $k_{i,j-IEPOX}^{-4}$ is more appropriate for predicting aerosol heterogeneous reactivity toward IEPOX
than other four sets of kinetic parameters.

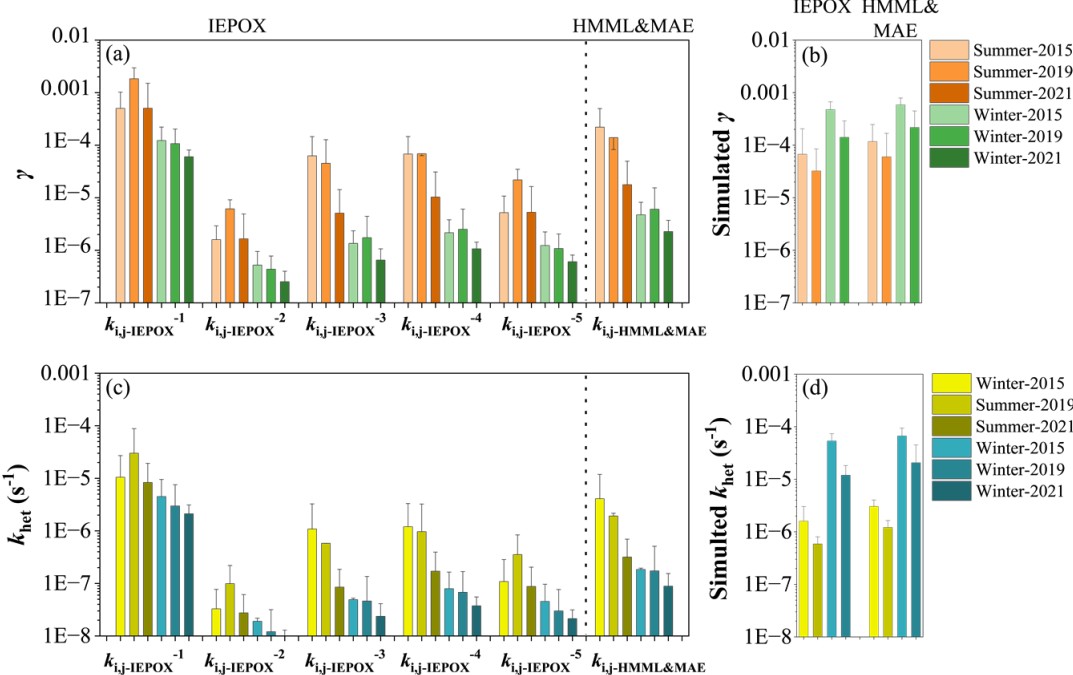


Figure 5. Reactive uptake coefficients (γ) and pseudo-first-order heterogeneous reaction rate constant
($k_{het}$, s$^{-1}$) of gas-phase IEPOX and HMML&MAE estimated using different sets of $k_{i,j-IEPOX}$ and $k_{i,j-}$
$_{HMML\&MAE}$ listed in Table 1 (a and c) and simulated by CMAQ model in Case 1 (b and d).

As shown in Figure 5a and 5c, the estimated γ and $k_{het}$ of IEPOX and HMML&MAE using $k_{i,j-}$

$_{IEPOX}^{-4}$ and $k_{i,j-HMML\&MAE}$ showed a similar trend which decreased in both winter and summer. Similar
decreasing trends in γ and $k_{het}$ were also simulated by CMAQ (Figure 5b and 5d). The calculated γ and
$k_{het}$ values of IEPOX and HMML&MAE in summer using $k_{i,j-IEPOX}^{-4}$ and $k_{i,j--HMML\&MAE}$, respectively, are
consistent with the simulated results, while the calculated values in winter were significantly lower than
the simulations, which is likely attributed to the significantly under-predicted aerosol pH and thereby
over-predicted aerosol reactivity in winter by the model. Overall, the declining trend of both calculated
and model-predicted $k_{het}$ offers an explanation for the decreasing trend of 2-MT-OS and 2-MG-OS in
summer, but it could not explain the observed trend of 2-MG and 2-MTs in summer.
**3.5 Meteorological influences on iSOA variation**
As discussed above, the variations in chemical factors (including atmospheric oxidizing capacity
and aerosol heterogeneous reactivity) could well explain the declining trend of both summertime and
wintertime iSOA OSs, but not the trend of summertime 2-MG and 2-MTs during the period of 2015-
2021. Since meteorological conditions could exert a significant influence on the concentrations of
atmospheric pollutants (Liu et al., 2023b; Gu et al., 2023), we further investigate the impact of the
variation in meteorological conditions on the variation of iSOA during the observation period.
To do so, the CMAQ simulations for 2019 adopted the emissions of 2015 (Test Case), so the
variations in simulated iSOA concentration from 2015 to 2019 in this case is mainly attributed to the
changes in the meteorological conditions during these years. As shown in Figure 6, the simulated
variations in median concentrations of 2-MTs, 2-MT-OS, 2-MG, and 2-MG-OS in the Test case
accounted for 38.3-82.4% and 66.5-99.2% of the concentration reductions in Case 1 in summer and
winter, respectively. This suggests that the alteration in meteorological conditions exerts a more
substantial influence on the variation in iSOA concentrations compared to the changes in emissions.

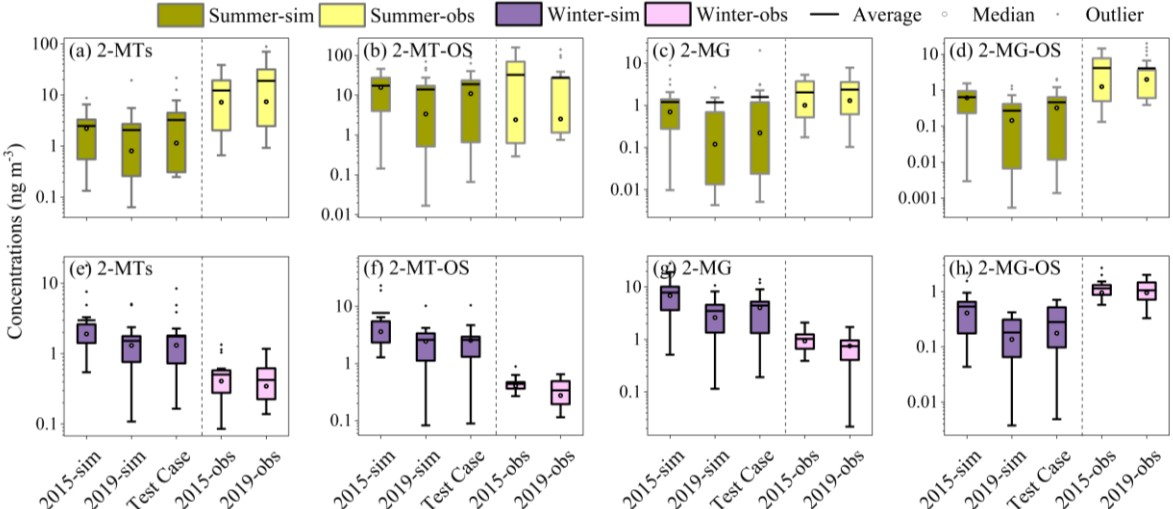


Figure 6. Simulated concentrations of 2-MTs, 2-MT-OS, 2-MG and 2-MG-OS in summer (a-d) and
winter (e-h) in Case 1 (2015-sim and 2019-sim) and Test Case (simulations with 2015 emissions and
2019 meteorological conditions). The observed concentrations of 2-MTs, 2-MT-OS, 2-MG, and 2-
MG-OS in 2015 and 2019 are also displayed (Detailed model-measurement comparisons are provided
in Section 3.2; after accounting for matrix effect, 2-MT-OS would decrease more sharply and 2-MG-
OS would show a descending inter-annual trend, consistent with model simulations).
It should be noted that the variations in meteorological conditions not only affect the physical
processes such as dilution and transport, but also influence the chemical processes determining the
formation of iSOA. To investigate the impacts of physical and chemical factors associated with the
variations in meteorological conditions on the abundance of iSOA, the concentrations of elemental
carbon (EC) were simulated in the Test Case. As shown in Figure 7, the simulated median
concentrations of summertime and wintertime EC in 2019 both decreased by 32.8%, respectively,
compared to those in 2015. Since EC is a primary pollutant and chemically inert under atmospheric
conditions, the variations in its concentration in the Test Case is attributed to the changes in the physical
processes, contributing approximately 59.5% of the EC reductions between 2015 and 2019 (Case 1).
As a result, we could expect a similar contribution of the physical processes to the reduction in iSOA
concentration. Notably, such reductions are significantly smaller than the simulated concentration
reductions of iSOA in the Test case (Figure 6), suggesting that the chemical factors associated with the
changes in meteorological conditions play a crucial role in determining the trend of iSOA in Shanghai,
consistent with the above analysis based on the observations. However, we note that the alternation in
meteorological conditions cannot explain the observed non-declining trend of 2-MG and 2-MTs in
summer, implying that some other factors that are not well represented in the CMAQ model may play
a role in controlling such a trend.

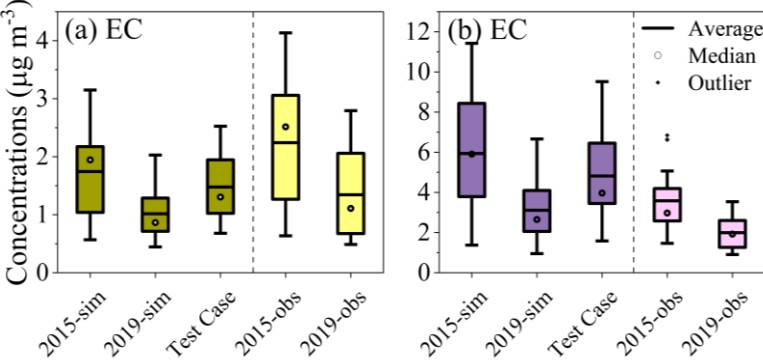


Figure 7. Simulated EC concentrations in Case 1 (2015-sim and 2019-sim) and Test Case (simulations
with 2015 emissions and 2019 meteorological conditions) in (a) summer and (b) winter. The measured
concentrations are also displayed and their trend is consistent with the simulations.

As discussed in Section 3.2, there is a large contribution of non-IEPOX sources to 2-MTs. Such

sources might not be well represented in the model, which could explain the discrepancy between the
simulated and observed trend of 2-MTs in summer. On the other hand, MACR is known as a first-
generation oxidation product of isoprene and an important intermediate for iSOA formation through
NOx-dominant pathways (Nguyen et al., 2015), but it can also originate from primary sources, such as
biological emissions (Jardine et al., 2012), residential wood burning (Gaeggeler et al., 2008), and
vehicle exhaust emissions (He et al., 2009). In certain urban areas, MACR arises primarily from
vehicular emissions, as observed at the Heshan site in the Pearl River Delta region, China (Ling et al.,
2019a) and in Houston, US (Park et al., 2011). In the present work, the simulated MACR concentration
demonstrated a decreasing trend from 2015 to 2019 in summer when the contribution of primary
emissions was considered in the model. Yet, our recent study revealed that, despite a good agreement
between modelled and measured isoprene concentrations, the model under-predicted the peak
concentrations of MACR during the noon in urban Shanghai (Li et al., 2022), implying an
underestimation of secondary formation and/or primary emissions of MACR in the model. Furthermore,
Gu et al. (2023) reported that in the southern cities of Jiangsu Province, which is adjacent to Shanghai,

anthropogenic VOCs increased by approximately 15% from 2015 to 2019. Therefore, we infer that the sources (e.g., primary emissions) of MACR might be under-predicted in this work, which may provide a rationale for the inconsistency between simulated and observed inter-annual trends of 2-MG.

**4. Conclusions**

In this study, observations of isoprene-derived SOA species in ambient $PM_{2.5}$ were conducted at an urban site in Shanghai, China during summers and winters in 2015, 2019, and 2021, aiming to understand the response of biogenic SOA formation to anthropogenic emission reductions in polluted regions. The complementary CMAQ model simulations were also performed for 2015 and 2019 and the results are compared to the measurements. It is found that the particulate concentration of total iSOA tracers, dominated by IEPOX-SOA species including 2-MT-OS and 2-MT, had a decreasing trend from 2015 to 2021 (55.6, 51.0, and 29.7 ng $m^{-3}$ in 2015, 2019, and 2021, respectively). When the measurement uncertainties of iSOA tracers such as the matrix effect of OSs were considered, 2-MT-OS and 2-MG-OS exhibited a declining trend in both seasons during the observation period. In contrast, 2-MG and 2-MTs showed no significant inter-annual variations After accounting for the gas-phase fraction, 2-MG even exhibited a slight upward trend in summer during these years.

The isoprene-derived SOA species correlated well with ozone and $O_x$ in summer but with sulfate in winter, suggesting that the atmospheric oxidation of isoprene to epoxide intermediates and their subsequent reactive uptake on aqueous aerosols are the key steps driving the formation of iSOA in summer and winter, respectively. The Ox-represented atmospheric oxidizing capacity and aerosol heterogeneous reactivity decreased significantly during the observation period, which provided an explanation for the decreasing trend of iSOA tracers in summer and winter (with the exception of summertime 2-MG and 2-MTs).

The CMAQ model predicted the concentrations of iSOA tracers reasonably well and captured the declining trend of total iSOA, 2-MT-OS, and 2-MG-OS in both seasons, but not the insignificant inter-annual variations of 2-MG and 2-MTs in summer. Further model simulations show that inter-annual variations in iSOA concentration are mainly governed by the changes in the meteorological conditions rather than the emissions. Consistent with the analysis based on the observation data, the model simulations show that the changes in chemical factors such as aerosol heterogeneous reactivity caused

by variations in meteorological conditions play an important role in controlling the inter-annual trend
of iSOA. The discrepancy between measured and modeled inter-annual trends of 2-MG and 2-MTs is
likely ascribed to the presence of unaccounted or underrepresented factors such as the direct emissions
of MACR and 2-MTs in the model. Overall, our study revealed the responses and underlying driving
factors of iSOA formation under rapidly changing anthropogenic emissions conditions in a typical
Chinese megacity. It also highlights the importance of tailored emission reductions for the mitigation
of PM formation from biogenic emissions through the regulation of atmospheric oxidizing capacity and
aerosol chemical reactivity.
**Data availability.**
The data presented in this work are available upon request from the corresponding author.
**Competing interests.**
The authors declare no competing interest relevant to this study.
**Author Contributions.**
YZ conceived and designed the study, HH, YW, and TY performed the field observation and analyzed
the data, JL, YL, TL, and YS performed the model simulations, HH, YZ, and JL wrote the paper, and
all other authors contributed to the discussion and writing.
**Acknowledgements.**
This study was supported by the National Key R&D Program of China (no. 2022YFC3701003) and the
National Natural Science Foundation of China (no. 22376137 and 22206120).

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
