# Peer review of "Pathway-specific responses of isoprene-derived secondary organic aerosol"

_EGUsphere, 2025_

## Author Comment (AC2)

We are grateful to the reviewer for the thoughtful comments, which are very helpful for improving our manuscript. Our point-to-point responses to each comment are as follows (the reviewer's comments are in black text, our responses are in blue text, and revised texts that appear in the manuscript are in red text).

General Comments:

The manuscript by Hu et al. presents interesting new results on isoprene-derived SOA in a megacity in China and the response to the on-going reductions in anthropogenic emissions. In general, the results are well presented and discussed. The study includes both measurements and modelling results. One limitation, which the authors should keep in mind, is that the field measurements are conducted on a limited number of samples. The modelling approach balances this, but the text sometimes overstates the conclusions that can be drawn.

Specific comments:

Line 21: profoundly is a somewhat strange word to use here.

**Response**: We have replaced "profoundly" with "strongly".

L28: Mass spectrometry – please be more specific.

**Response:** We have specified the type of mass spectrometry in the text.

L26-27: "…were measured by gas chromatography-mass spectrometry and high-resolution liquid chromatography-mass spectrometry."

L38: "regulating atmospheric oxidizing capacity and aerosol reactivity to mitigate biogenic SOA formation" Is biogenic SOA formation really the major source of air pollution? How can atmospheric oxidizing capacity and aerosol reactivity be regulated?

Response: Thanks for the reviewer's comment. The contribution of biogenic SOA to ambient particulate matter varies significantly across different locations and seasons. For example, observational studies revealed that biogenic SOA contributed to approximately half of total fine organic aerosol in southeastern US (Liao et al., 2007; Zhang et al., 2018). Modelling studies found that biogenic sources accounted for 61% of SOA on average in China, with its proportion reaching 75% in summer but dropping to 24% in winter (Hu et al., 2017). In addition, the proportion of biogenic SOA increased with anthropogenic emission reductions in recent years (Zhang et al., 2024). These results suggest an important contribution of biogenic SOA to PM pollution.

The atmospheric oxidizing capacity is largely governed by the concentrations of oxidants including hydroxyl radicals, ozone, and nitrate radicals, which can be regulated by reducing anthropogenic emissions such as VOCs and NOx (Fiore et al., 2024; Dai et al., 2024). The aerosol reactivity is controlled by its concentration and composition, which can be regulated by reducing anthropogenic emissions such as VOCs, NOx, and $SO_2$.

In the revised manuscript, this sentence has been revised as follows:

Line 36-39: "These findings highlight pathway-specific iSOA responses to emission reductions in a megacity and the importance of targeted anthropogenic emission reductions for mitigating biogenic SOA formation through regulating atmospheric oxidizing capacity and aerosol reactivity."

L46: Heald et al., 2008 -is there a newer reference.

Response: Thanks. The reference has been updated here as follows:

L44-46: "…the oxidation of isoprene contributes significantly to the formation of secondary organic aerosol (SOA) in the troposphere, estimated to 19.6 TgC $yr^{-1}$ (Kelly et al., 2018)."

L58-59: "The gaseous IEPOX can be taken up into aqueous aerosol and undergo acid-catalyzed reactions to form polyols, organosulfates (OSs), and oligomers." This sentence needs references.

Response: Thanks. We have added references to this sentence.

L57-60: "The gaseous IEPOX can be taken up into aqueous aerosol and undergo acid-catalyzed reactions to form polyols, organosulfates (OSs), and oligomers (Hettiyadura et al., 2019; Lin et al., 2012; Surratt et al., 2010)."

L61: "abundantly measured" it is unclear what you mean – where they measured a lot or in high concentrations?

Response: We want to mean here the IEPOX-SOA tracers were measured a lot. To be more precise, we have replaced "abundantly" with "extensively".

L67: The sign before 45% is missing.

Response: Thanks. The sign before 45% is "~", and we have added it.

L100: Which iSOA tracers?

Response: The iSOA tracers here include polyols tracers such as 2-MTs and 2-MG. We have replaced the sentence here with a clearer statement.

Lines 98-102: "Furthermore, while the iSOA polyol tracers are formed in the particle phase, they can actively partition between gas and particle phases due to their semi-volatile characteristics (Fan et al., 2020; Isaacman-Vanwertz et al., 2016; Nguyen et al., 2015). As a result, considering their particle-phase concentration only may bias our understanding of the atmospheric abundance and chemistry of iSOA."

L108: cation ions -> cations

Response: We have revised this.

L109: The text is about recent changes but one of the references is from 2014.

Response: We have updated the reference to newer ones.

L104-107: "For example, $PM_{2.5}$ mass concentrations dropped significantly, with a marked reduction in water-soluble inorganic ions such as sulfate, ammonium, and cation ions (Liu et al., 2023a; Zheng et al., 2018; Liu et al., 2023b; Geng et al., 2024)…"

L113: later -> latter

Response: We have corrected this typo.

L119: comprehend -> understand

Response: We have revised this.

L149. What is KPA?

Response: KPA refers to ketopinic acid that has been widely used as a surrogate standard in the quantification of iSOA polyol tracers. We have replaced KPA with its full name ketopinic acid.

L151. Please provide information about sources of authentic standards.

Response: Thanks. We have added the sources of the standards.

L149-150: "…2-MG and 2-methylerythritol were quantified using their authentic standards (Toronto Research Chemicals, 99.8%)."

L171-172: Can this depend on the vendor of the ESI inlet and MS?

Response: Yes, the ESI efficiency is affected by various factors, including the type and configuration of the ESI inlet and MS as well as their specific operating conditions. Considering that the ESI efficiency obtained from one instrument may not be directly applied to another instrument, we have deleted the

discussion in lines 171-172.

L181: Please be a bit more specific about the distance and the surrounding areas i.e. emissions.

Response: We have added a more detailed description about this.

L169-173: "Temperature, relative humidity (RH), as well as the concentrations of trace gases and $PM_{2.5}$ were measured at a state-controlled air quality monitoring station on the Xuhui Campus of Shanghai Normal University, which is surrounded by residential areas and commercial districts and 4.5 km southwest of the $PM_{2.5}$ sampling site of this work."

L194-195: Please discuss the uncertainty associated with this approach.

Response: Previous studies have found that lacking gas-phase inputs of ammonia could lead to underprediction of pH using thermodynamic equilibrium models, such as ISORROPIA and E-AIM (Hennigan et al., 2015; Guo et al., 2015; Song et al., 2018). Guo et al. (2015) found that the pH values were underestimated by 1 unit on average in southeast U.S. when using only aerosol ammonium data as inputs in ISORROPIA model. Similarly, Song et al. (2018) found that a 10-fold increase in gas-phase $NH_3$ concentrations roughly corresponds to a 1 unit increase in pH in the ammonia-rich atmosphere like Beijing. In addition, we found that the average difference between aerosol pH predicted by using aerosol ammonium only and gas-plus-particle-phase ammonia as inputs was about 1 unit in 2015 and 2019. Thus, we inferred that the lack of gas-phase concentrations of ammonia might lead to underestimation of pH by ~1 unit and increased the output pH estimated using aerosol ammonium only as input by one unit in 2021 to reduce the uncertainty arising from lack of gas-phase $NH_3$ in pH estimation.

We have added the following discussion to the revised manuscript.

L182-195: "Additionally, the 2021 pH was predicted by the ISORROPIA-II using particle-phase-only concentrations of ions as input due to lack of gas-phase $NH_3$ data. Previous studies have found that lacking gas-phase inputs of ammonia could lead to under-prediction of pH using thermodynamic equilibrium models, such as ISORROPIA and E-AIM (Hennigan et al., 2015a; Guo et al., 2015; Song et al., 2018). Guo et al. (2015) found that the pH values were underestimated by 1 unit on average in southeast US when using only aerosol ammonium data as inputs in ISORROPIA model. Similarly, Song et al. (2018) found that a 10-fold increase in gas-phase $NH_3$ concentrations roughly corresponds to a 1 unit increase in pH in the ammonia-rich atmosphere like Beijing. In addition, we found that aerosol pH in 2015 and 2019 predicted using aerosol ammonium only as input in our study was on average 1 unit lower than that predicted using gas-plus-particle-phase ammonia as input in Zhou et al. (2022). Thus, we inferred that the lack of gas-phase concentrations of ammonia might lead to underestimation of pH by ~1 unit in the present study and increased the output pH estimated using aerosol ammonium only as input by one unit to represent aerosol acidity in 2021."

L224: It is nice that the authors bring forward this uncertainty, but it is unclear how it affects the results presented here and whether it should have been tested if the magnitude of the effect is the same in the current study.

Response: Thanks for the reviewer's comment. Reviewer #1 has also raised this concern (comment #1) and we have addressed it in detail in our responses to his/her comments.

The matrix effect factors and relative ionization efficiencies reported by Bryant et al. (2021) may not be valid for the present study, given that these factors could vary significantly across studies due to the differences in ambient samples analyzed and in analytical instruments and specific instrumental conditions. Therefore, we have conducted a set of experiments to evaluate the matrix effect of iSOA OSs and the uncertainties in the abundance, trend, and relative ratios of different iSOA tracers in the present study and added relevant discussions to the revised manuscript (see below).

Lines 221-255: "Previous studies have demonstrated that the concentrations of 2-MT-OS and 2-MG-OS were significantly underestimated due to matrix effect by using reversed phase liquid chromatography-mass spectrometry (RPLC-MS) (Hettiyadura et al., 2015; Bryant et al., 2020; Bryant et al., 2021). In the present work, because of lack of authentic standards of isoprene-derived OSs, we are not able to quantify the absolute value of underestimation in the concentration of 2-MT-OS and 2-MG-OS due to the matrix effect. However, using ambient $PM_{2.5}$ samples with different concentrations, we can quantify the relative extent of underestimation in OS concentrations due to matrix effect in different samples, which allows for an evaluation of uncertainties in the abundance, trend, and relative ratios of different iSOA tracers in

this study. In the matrix effect experiments, the extracts of ambient $PM_{2.5}$ samples with different concentrations were mixed and the measured signals of 2-MT-OS and 2-MG-OS in mixed extracts were compared to the sum of OS signals detected separately in individual extracts. The concentrations of $PM_{2.5}$, sulfate, as well as 2-MT-OS and 2-MG-OS in ambient samples used for this evaluation are listed in Table S3. The relative matrix effect factor ($F_{matrix}$), defined as the ratio of the measured OS signals in mixed extracts to the sum of OS signals measured in each extract before mixing, are used to evaluate the matrix effect of OSs. A $F_{matrix}$ value of less than 1 indicates the presence of matrix effect.

As shown in Figure S2, the $F_{matrix}$ values were significantly smaller than 1 in both summer and winter, indicating that the signal responses of 2-MT-OS and 2-MG-OS in mixed extracts were largely suppressed due to the matrix effect. Notably, $F_{matrix}$ exhibits a significant negative dependence on the reduced mass ($\mu$, $\mu g\ m^{-3}$), a proxy used to represent effective mass loadings of the mixed $PM_{2.5}$ extracts, defined as:

$$\mu = \sqrt{m_1 * m_2/(m_1 + m_2)} \qquad (1)$$

where $m_1$ and $m_2$ are the $PM_{2.5}$ mass loading of individual samples. This observation suggests that the concentrations of iSOA OSs in $PM_{2.5}$ samples collected in 2015 were underestimated more than those in 2021, given that ambient $PM_{2.5}$ concentrations declined by 39.8% and 47.0% from 2015 to 2021 in summer and winter, respectively.

As $PM_{2.5}$ concentrations in ambient samples used for the matrix effect evaluation generally represent lower or upper ends of $PM_{2.5}$ concentrations during the observation period (see Table S3), the relative differences in measured $F_{matrix}$ values at varying reduced mass (Figure S3) may roughly reflect the differences in the extent of underestimation in OS concentrations for samples collected across 2015-2021. During summer, the $F_{matrix}$ value decreased from 0.71 to 0.63 for 2-MT-OS and from 0.85 to 0.58 for 2-MG-OS with increasing reduced mass, indicating that the concentrations of these two iSOA OSs were a factor of 1.2 and 1.5 more underestimated in 2015 than in 2021 due to matrix effect. Similarly, during winter the $F_{matrix}$ values of iSOA OSs decreased from 0.9 to 0.6 with rising reduced mass, implying a factor of 1.5 greater underestimation in OS concentrations in 2015 than in 2021."

Lines 371-379: "However, the inter-annual trend of iSOA OSs could be altered due to the matrix effect. The measured concentration of 2-MT-OS exhibited a decreasing inter-annual trend, while 2-MG-OS showed insignificant variation between 2015-2021. Accounting for the significantly larger matrix effects in 2015 samples compared to 2021 samples (see Section 2.3), the true concentrations of 2-MT-OS would decrease more sharply and 2-MG-OS might also exhibit a declining trend."

Lines 411-413: "However, given the significant underestimation of iSOA OSs due to matrix effect and overestimation of $C_5$-alkene triols due to their potential artifact formation, the true concentration of iSOA OSs would predominate over that of polyol tracers."

Table S3. Major components in eight $PM_{2.5}$ filter samples that used for estimating matrix effect of 2-MT-OS and 2-MG-OS.

| Season | Sample | Sampling date | $PM_{2.5}$ ($\mu g\ m^{-3}$) | $SO_4^{2-}$ ($\mu g\ m^{-3}$) | 2-MG-OS ($ng\ m^{-3}$) | 2-MT-OS ($ng\ m^{-3}$) |
|--------|--------|---------------|------|------|---------|---------|
| Summer | High 1 | 2015/7/28 | 35.66 | 7.74 | 13.04 | 157.89 |
|  | High 2 | 2019/7/24 | 31.76 | 4.58 | 12.36 | 100.61 |
|  | Low 1 | 2019/8/5 | 10.04 | 2.29 | 0.58 | 0.83 |
|  | Low 2 | 2021/8/5 | 12.21 | 1.48 | 0.18 | 0.18 |
| Winter | High 3 | 2022/1/2 | 77.76 | 6.28 | 1.80 | 0.47 |
|  | High 4 | 2022/1/3 | 65.38 | 5.41 | 1.54 | 0.41 |
|  | Low 3 | 2021/12/25 | 18.04 | 3.42 | 0.49 | 0.07 |
|  | Low 4 | 2021/12/26 | 23.71 | 3.01 | 0.70 | 0.08 |

[Figure]

Figure S2. Relative matrix effect factors (defined as the ratio of signal response in mixed extracts to the sum of signal response in individual extracts, $F_{matrix}$) of 2-MT-OS and 2-MG-OS under different mixture type in summer (a) and winter (b) (yellow background represents low-plus-low mixture, green background represents high-plus-low mixture, and blue background represents high-plus-high mixture).

L299: Are the values statistically significantly different?

Response: Results from the t significance test indicated no statistically significant differences in ozone concentrations among the years 2015, 2019, and 2021, though the average values of MDA $O_3$ had a slight decrease.

L302: Dramatic is not a correct scientific word to use here. It would be nice if the authors could list some numbers.

Response: Thanks. The word "dramatic" was replaced by "drastic", and the proportion of major composition of $PM_{2.5}$ have been added there.

L331-333: "During the observation period, $PM_{2.5}$ concentrations decreased by 56.5%, with concentrations of major components, including sulfate, ammonium, and OM, decreasing by 51.8%, 40.6%, and 39.1%, respectively (see Figure 1)"

L303: Are there other types of nitrate than aerosol nitrate?

Response: There are no other types of nitrate than aerosol nitrate. For simplicity, we have replaced "aerosol nitrate" with "nitrate".

L305: Please also state the average values.

Response: Thanks, we have added the average values for the fraction.

L335-337: "Overall, OM was the most abundant component in $PM_{2.5}$, accounting for 10.2-72.7% (average 22.6%) of total $PM_{2.5}$ mass, followed by sulfate (6.8-45.2%, average 17.7%), nitrate (0.5-32.6%, average 17.0%), and ammonium (1.1-18.2%, average 8.8%)."

L308: Please avoid the use of the word dramatic. Aerosol pH – are the differences statistically significantly different?

Response: Thanks. We have conducted a t-test for aerosol pH and LWC, and the result shows that the differences in aerosol pH and LWC across different years are significantly different (p < 0.05).

We have modified the sentence as follows:

L337-340: "Ascribed to the strong decrease of inorganic ion concentrations (in particular sulfate), aerosol LWC decreased from $9.14 \pm 4.51$ µg m$^{-3}$ in 2015 to $4.40 \pm 2.76$ µg m$^{-3}$ in 2021 (p < 0.05). Aerosol pH decreased from $3.2 \pm 0.4$ in 2015 to $2.5 \pm 0.9$ in 2021 (p < 0.05)…"

Figure 1: The figure is very small and it is difficult to see details.

Response: We have updated Figure 1 to a clearer version.

[Figure]

L334: Please state standard deviations.

Response: Thanks. We have included the standard deviations for the concentration values.

L408-409: "The annual average particulate concentration of the total iSOA polyol tracers (including 2-MTs, 2-MG, and C$_5$-alkene triols) were $36.1 \pm 63.3$, $33.4 \pm 70.5$, and $18.7 \pm 47.1$ ng m$^{-3}$ in 2015, 2019 and 2021…"

L361: What do you mean by median?

Response: We have replaced "median" with "moderate".

L365-367: Please clarify this sentence.

Response: We have deleted this sentence since we have added detailed discussion about the modeled result of individual iSOA tracers in the revised manuscript.

L368: Is it consistent or different?

Response: Thanks. There is a typo here and we have deleted the word "in" to emphasize the consistence between the model simulation results in our study and previous studies.

L396-398: It seems like a bold statement to say that the concentrations in winter decrease when the values are $0.8 \pm 0.3$, $0.8 \pm 0.2$, and $0.7 \pm 0.4$.

Response: We have deleted this statement since the large uncertainties arising from thermal degradation of $C_5$-alkene triols and the matrix effect of iSOA OSs would make it difficult to discuss the inter-annual trend of the IEPOX-SOA/HMML&MAE-SOA ratio.

L400-403: These sentences need clarification.

Response: Thanks. These sentences have been deleted since large uncertainties existing in quantification of $C_5$-alkene triols makes it difficult to discuss the trend of the ratios of IEPOX-SOA/HMML&MAE-SOA.

Figure 4: It is not possible to read the numbers in the heatmap.

Response: Thanks. The heatmap has been replaced with a clearer version.

[Figure]

Figure 4. Coefficients of correlation ($r^2$) between iSOA compounds and various influencing factors of iSOA formation in (a-c) summer and (d-f) winter of 2015, 2019, and 2021, respectively. The ISOP is the abbreviation of isoprene.

L429-431: The correlation is quite low, which should be reflected better in the text.

Response: We have used a more neutral description for the correlation, replacing "moderate" with "weak to moderate".

L434-436: The correlation is quite low to moderate, which should be reflected better in the text.

Response: We have replaced "well" with "moderately".

L439: The change is quite small. Was a statistical test performed to check this?

Response: A t-test was performed to evaluate the statistical significance of the changes in wintertime IEPOX-SOA, LWC, and sulfate concentrations across different years. The result shows that the changes are statistically significant ($p < 0.05$) during the observation period.

L451: What happens to isoprene when the oxidation capacity is lower? Does it not just take longer for the oxidation to occur?

Response: Yes, the lower oxidation capacity generally means lower level of oxidants, thus the oxidation of isoprene would be slower, which could lead to decreased formation of iSOA.

L496-498: Unclear sentence.

Response: The sentence has been rephrased as follows:

L564-566: "It also suggests that the $k_{i,j\text{-IEPOX}}^{-4}$ is more appropriate for predicting aerosol heterogeneous reactivity toward IEPOX than other four sets of kinetic parameters."

Figure 6: Describe better what the numbers/years for the cases are.

Response: The x-axis label in Figure 6 has been revised and its meaning is explained in the figure caption. In addition, we have added the measured data to this figure according to Reviewer #1's suggestion.

[Figure]

Figure 6. Simulated concentrations of 2-MTs, 2-MT-OS, 2-MG and 2-MG-OS in summer (a-d) and winter (e-h) in Case 1 (2015-sim and 2019-sim) and Test Case (simulations with 2015 emissions and 2019 meteorological conditions). The observed concentrations of 2-MTs, 2-MT-OS, 2-MG, and 2-MG-OS in 2015 and 2019 are also displayed (Detailed model-measurement comparisons are provided in Section 3.2; after accounting for matrix effect, 2-MT-OS would decrease more sharply and 2-MG-OS would show a descending inter-annual trend, consistent with model simulations).

L551: This line needs editing.

Response: Thanks. This sentence has been revised as follows:

L626-630: "On the other hand, MACR is known as a first-generation oxidation product of isoprene and an important intermediate for iSOA formation through NOx-dominant pathways (Nguyen et al., 2015), but it can also originate from primary sources, such as biological emissions (Jardine et al., 2012), residential wood burning (Gaeggeler et al., 2008), and vehicle exhaust emissions (He et al., 2009)"

L554: "Field studies have demonstrated" – this is a bold statement as there are differences between urban areas.

Response: This statement has been revised to improve clarity.

L630-632: "In certain urban areas, MACR arises primarily from vehicular emissions, as observed at the Heshan site in the Pearl River Delta region, China (Ling et al., 2019a) and in Houston, US (Park et al., 2011). "

L564: underappreciated is a loaded word not suitable here.

Response: We have replaced the word "underappreciated" with "underpredicted".

L567: three- year measurement gives the impression that samples for three complete years were studied. Please correct.

Response: We have replaced "a three-year measurement" with "an observation" and added "during the summer and winter in 2015, 2019, and 2021" to the end of this sentence to more accurately reflect the temporal coverage of the data.

L568: During the period 2015-2021, however only periods during three years were studied. Please correct.

Response: We have replaced "during the period 2015-2021" with "during the summer and winter in 2015, 2019, and 2021."

L574-575: "while HMML/MAE-SOA species (4.3, 4.2, and 4.3 ng m$^{-3}$ in 2015, 2019, and 2021, respectively) did not decrease significantly". No they did not seem to decrease at all! Please correct.

Response: Thanks. We have deleted the word "significantly" in this sentence.

L579-581: Please check that this can be concluded based on the current study.

Response: Thanks. The correlation analysis showed that iSOA exhibited a good correlation with O$_3$ and Ox in summer and sulfate in winter. This indicates that the level of atmospheric oxidants (that largely determines the rate of isoprene oxidation) and sulfate aerosol (that plays a crucial role in the reactive uptake of isoprene-derived epoxide intermediates) are key factors driving iSOA formation in summer and winter, respectively. Based on this, we can conclude that "The atmospheric oxidation of isoprene to epoxide intermediates and their subsequent reactive uptake on aqueous aerosols are the key steps driving the formation of iSOA in summer and winter, respectively."

L582-583: This sentence is not clear.

Response: This sentence has been revised as follows:

L657-658: "The Ox-represented atmospheric oxidizing capacity and aerosol heterogeneous reactivity decreased significantly during the observation period …"

L599: Are biogenic emissions pollution?

Response: We have modified "PM pollution from biogenic emissions" as "PM formation from biogenic emissions".

References:

Bryant, D. J., Elzein, A., Newland, M., White, E., Swift, S., Watkins, A., Deng, W., Song, W., Wang, S., Zhang, Y., Wang, X., Rickard, A. R., and Hamilton, J. F.: Importance of Oxidants and Temperature in the Formation of Biogenic Organosulfates and Nitrooxy Organosulfates, ACS Earth Space Chem., 5, 2291-2306, 10.1021/acsearthspacechem.1c00204, 2021.

Bryant, D. J., Dixon, W. J., Hopkins, J. R., Dunmore, R. E., Pereira, K. L., Shaw, M., Squires, F. A., Bannan, T. J., Mehra, A., Worrall, S. D., Bacak, A., Coe, H., Percival, C. J., Whalley, L. K., Heard, D. E., Slater, E. J., Ouyang, B., Cui, T., Surratt, J. D., Liu, D., Shi, Z., Harrison, R., Sun, Y., Xu, W., Lewis, A. C., Lee, J. D., Rickard, A. R., and Hamilton, J. F.: Strong anthropogenic control of secondary organic aerosol formation from isoprene in Beijing, Atmos. Chem. Phys., 20, 7531-7552, 10.5194/acp-20-7531-2020, 2020.

Dai, J., Brasseur, G. P., Vrekoussis, M., Kanakidou, M., Qu, K., Zhang, Y., Zhang, H., and Wang, T.: The atmospheric oxidizing capacity in China – Part 2: Sensitivity to emissions of primary pollutants, Atmos. Chem. Phys., 24, 12943-12962, 10.5194/acp-24-12943-2024, 2024.

Fiore, A. M., Mickley, L. J., Zhu, Q., and Baublitz, C. B.: Climate and Tropospheric Oxidizing Capacity, Annu. Rev. Earth. Pl. Sc., 52, 321-349, https://doi.org/10.1146/annurev-earth-032320-090307, 2024.

Gaeggeler, K., Prevot, A. S. H., Dommen, J., Legreid, G., Reimann, S., and Baltensperger, U.: Residential wood burning in an Alpine valley as a source for oxygenated volatile organic compounds, hydrocarbons and organic acids, Atmos. Environ., 42, 8278-8287, 10.1016/j.atmosenv.2008.07.038, 2008.

Geng, G., Liu, Y., Liu, Y., Liu, S., Cheng, J., Yan, L., Wu, N., Hu, H., Tong, D., Zheng, B., Yin, Z., He, K., and Zhang, Q.: Efficacy of China's clean air actions to tackle PM2.5 pollution between 2013 and 2020, Nat. Geosci., 17, 987-994, 10.1038/s41561-024-01540-z, 2024.

Guo, H., Xu, L., Bougiatioti, A., Cerully, K. M., Capps, S. L., Hite Jr, J. R., Carlton, A. G., Lee, S. H., Bergin, M. H., Ng, N. L., Nenes, A., and Weber, R. J.: Fine-particle water and pH in the southeastern United States, Atmos. Chem. Phys., 15, 5211-5228, 10.5194/acp-15-5211-2015, 2015.

He, C., Ge, Y. S., Tan, J. W., You, K. W., Han, X. K., Wang, J. F., You, Q. W., and Shah, A. N.: Comparison of carbonyl compounds emissions from diesel engine fueled with biodiesel and diesel, Atmos. Environ., 43, 3657-3661, 10.1016/j.atmosenv.2009.04.007, 2009.

Hennigan, C. J., Izumi, J., Sullivan, A. P., Weber, R. J., and Nenes, A.: A critical evaluation of proxy methods used to estimate the acidity of atmospheric particles, Atmos. Chem. Phys., 15, 2775-2790, 10.5194/acp-15-2775-2015, 2015.

Hettiyadura, A. P. S., Al-Naiema, I. M., Hughes, D. D., Fang, T., and Stone, E. A.: Organosulfates in Atlanta, Georgia: anthropogenic influences on biogenic secondary organic aerosol formation, Atmos. Chem. Phys., 19, 3191-3206, 10.5194/acp-19-3191-2019, 2019.

Hu, J., Wang, P., Ying, Q., Zhang, H., Chen, J., Ge, X., Li, X., Jiang, J., Wang, S., Zhang, J., Zhao, Y., and Zhang, Y.: Modeling biogenic and anthropogenic secondary organic aerosol in China, Atmos. Chem. Phys., 17, 77-92, 10.5194/acp-17-77-2017, 2017.

Jardine, K. J., Monson, R. K., Abrell, L., Saleska, S. R., Arneth, A., Jardine, A., Ishida, F. Y., Serrano, A. M. Y., Artaxo, P., Karl, T., Fares, S., Goldstein, A., Loreto, F., and Huxman, T.: Within-plant isoprene oxidation confirmed by direct emissions of oxidation products methyl vinyl ketone and methacrolein, Global Change Biol., 18, 973-984, 10.1111/j.1365-2486.2011.02610.x, 2012.

Kelly, J. M., Doherty, R. M., O'Connor, F. M., and Mann, G. W.: The impact of biogenic, anthropogenic, and biomass burning volatile organic compound emissions on regional and seasonal variations in secondary organic aerosol, Atmos. Chem. Phys., 18, 7393-7422, 10.5194/acp-18-7393-2018, 2018.

Liang, S., Wang, Y., Chen, H., Chan, W., and Yu, J. Z.: Accurate Quantification of Multifunctional C2–3 Organosulfates in Atmospheric Aerosols Using Liquid Chromatography-Electrospray Ionization Mass Spectrometry: Overcoming Matrix Effects and Underestimation, Environ. Sci. Technol., 10.1021/acs.est.5c01846, 2025.

Liao, H., Henze, D. K., Seinfeld, J. H., Wu, S., and Mickley, L. J.: Biogenic secondary organic aerosol over the United States: Comparison of climatological simulations with observations, J. Geophys. Res.-Atmos., 112, https://doi.org/10.1029/2006JD007813, 2007.

Lin, Y. H., Zhang, Z. F., Docherty, K. S., Zhang, H. F., Budisulistiorini, S. H., Rubitschun, C. L., Shaw, S. L., Knipping, E. M., Edgerton, E. S., Kleindienst, T. E., Gold, A., and Surratt, J. D.: Isoprene Epoxydiols as Precursors to Secondary Organic Aerosol Formation: Acid-Catalyzed Reactive Uptake Studies with Authentic Compounds, Environ. Sci. Technol., 46, 250-258, 10.1021/es202554c, 2012.

Ling, Y., Wang, Y., Duan, J., Xie, X., Liu, Y., Peng, Y., Qiao, L., Cheng, T., Lou, S., Wang, H., Li, X., and Xing, X.: Long-term aerosol size distributions and the potential role of volatile organic compounds (VOCs) in new particle formation events in Shanghai, Atmos Environ, 202, 345-356, 10.1016/j.atmosenv.2019.01.018, 2019.

Liu, P., Ding, X., Li, B. X., Zhang, Y. Q., Bryant, D. J., and Wang, X. M.: Quality assurance and quality control of atmospheric organosulfates measured using hydrophilic interaction liquid chromatography (HILIC), Atmos. Meas. Tech., 17, 3067-3079, 10.5194/amt-17-3067-2024, 2024.

Liu, Y., Yang, X., Tan, J., and Li, M.: Concentration prediction and spatial origin analysis of criteria air pollutants in Shanghai, Environ. Pollut., 327, 121535, https://doi.org/10.1016/j.envpol.2023.121535, 2023a.

Liu, Y., Geng, G., Cheng, J., Liu, Y., Xiao, Q., Liu, L., Shi, Q., Tong, D., He, K., and Zhang, Q.: Drivers of Increasing Ozone during the Two Phases of Clean Air Actions in China 2013-2020, Environ. Sci. Technol., 57, 10.1021/acs.est.3c00054, 2023b.

Nguyen, T. B., Bates, K. H., Crounse, J. D., Schwantes, R. H., Zhang, X., Kjaergaard, H. G., Surratt, J. D., Lin, P., Laskin, A., Seinfeld, J. H., and Wennberg, P. O.: Mechanism of the hydroxyl radical oxidation of methacryloyl peroxynitrate (MPAN) and its pathway toward secondary organic aerosol formation in the atmosphere, Phys. Chem. Chem. Phys., 17, 17914-17926, 10.1039/c5cp02001h, 2015.

Park, C., Schade, G. W., and Boedeker, I.: Characteristics of the flux of isoprene and its oxidation products in an urban area, J. Geophys. Res.-Atmos., 116, 10.1029/2011jd015856, 2011.

Song, S., Gao, M., Xu, W., Shao, J., Shi, G., Wang, S., Wang, Y., Sun, Y., and McElroy, M. B.: Fine-particle pH for Beijing winter haze as inferred from different thermodynamic equilibrium models, Atmos. Chem. Phys., 18, 7423-7438, 10.5194/acp-18-7423-2018, 2018.

Surratt, J. D., Chan, A., Eddingsaas, N. C., Chan, M. N., Loza, C. L., Kwan, A. J., Hersey, S. P., Flagan, R. C., Wennberg, P. O., and Seinfeld, J. H.: Reactive intermediates revealed in secondary organic

aerosol formation from isoprene, Proc. Natl. Acad. Sci. U.S.A., 107, 6640-6645, 2010.

Zhang, H., Yee, L. D., Lee, B. H., Curtis, M. P., Worton, D. R., Isaacman-VanWertz, G., Offenberg, J. H., Lewandowski, M., Kleindienst, T. E., Beaver, M. R., Holder, A. L., Lonneman, W. A., Docherty, K. S., Jaoui, M., Pye, H. O. T., Hu, W., Day, D. A., Campuzano-Jost, P., Jimenez, J. L., Guo, H., Weber, R. J., de Gouw, J., Koss, A. R., Edgerton, E. S., Brune, W., Mohr, C., Lopez-Hilfiker, F. D., Lutz, A., Kreisberg, N. M., Spielman, S. R., Hering, S. V., Wilson, K. R., Thornton, J. A., and Goldstein, A. H.: Monoterpenes are the largest source of summertime organic aerosol in the southeastern United States, Proc. Natl. Acad. Sci. U.S.A., 115, 2038-2043, doi:10.1073/pnas.1717513115, 2018.

Zhang, W., Liu, Y., Yue, M., Dong, X., Huang, K., and Wang, M.: Understanding the long-term trend of organic aerosol and the influences from anthropogenic emission and regional climate change in China, Atmos. Chem. Phys., 25, 3857-3872, 10.5194/acp-25-3857-2025, 2024.

Zheng, B., Tong, D., Li, M., Liu, F., Hong, C., Geng, G., Li, H., Li, X., Peng, L., Qi, J., Yan, L., Zhang, Y., Zhao, H., Zheng, Y., He, K., and Zhang, Q.: Trends in China's anthropogenic emissions since 2010 as the consequence of clean air actions, Atmos. Chem. Phys., 18, 14095-14111, 10.5194/acp-18-14095-2018, 2018.

Zhou, M., Zheng, G., Wang, H., Qiao, L., Zhu, S., Huang, D., An, J., Lou, S., Tao, S., Wang, Q., Yan, R., Ma, Y., Chen, C., Cheng, Y., Su, H., and Huang, C.: Long-term trends and drivers of aerosol pH in eastern China, Atmos. Chem. Phys., 22, 13833-13844, 10.5194/acp-22-13833-2022, 2022.

---

## Author Response (AR1)

11 September 2025

Dear Professor Surratt,

Attached please find our revised manuscript entitled "Pathway-specific responses of isoprene-derived secondary organic aerosol formation to anthropogenic emission reductions in a megacity in eastern China". Also attached are a point-by-point response to your and reviewers' comments and a track-change version of the revised manuscript. We greatly appreciate your and reviewers' thoughtful comments and suggestions on our manuscript.

Major changes we have made in the manuscript are as follows.

1) We have conducted additional experiments to quantify the relative extent of underestimation in iSOA OS concentrations in samples collected across 2015-2021 due to the matrix effect during RPLC-MS analysis and evaluated the associated uncertainties in the abundance, trend, and relative ratios of different iSOA tracers.

2) We have added new discussions regarding the potential artifact formation of $C_5$-alkene triols during GC-MS analysis and the resulting uncertainties in the relative ratios of iSOA OSs to polyols tracers and IEPOX-SOA to HMML&MAE-SOA.

3) We have performed a sensitivity test to evaluate the influence of matrix effect on the correlation analysis of iSOA OSs with different influencing factors. The result demonstrates that the underestimation in OS concentrations did not change the conclusion of this analysis.

4) We have included a new set of reaction kinetic values, recommended by Chen et al. (2024, ACS ES&T Air), in the estimation of reactive uptake coefficients of IEPOX and aerosol heterogeneous reactivity.

5) We have provided more thorough comparisons of the measured concentrations and relative ratios of iSOA tracers with the modelled values in our study and urban observations in previous studies, and considered the measurement uncertainties of iSOA OSs and $C_5$-alkene triols in such comparisons.

We think our revised manuscript is much improved in terms of scientific and presentation quality.

We look forward to hearing from you regarding this manuscript.

Yours sincerely,

Yue Zhao
Professor
School of Environmental Science and Engineering
Shanghai Jiao Tong University
800 Dongchuan Road, Shanghai 200240, China

We sincerely appreciate the time and effort the editor has dedicated to evaluating our manuscript. The editor's comments are insightful, which will greatly help us improve the quality of this work. Our point-to-point responses to each comment are as follows (Comments #1 and #2 have already been addressed before the full review period of ACP).

Comments:

Dear authors, I would like to request a just a couple of edits before posting online for review and comments from the community:

1.) Can you add to the experimental section what kind of LC column you used? It sounds like from the paper you referenced that you used RPLC to measure and resolve organosulfates (OSs) from isoprene?

**Response:** Yes, we used RPLC to measure isoprene-derived OSs in our previous and current work. We have added a description of the type of LC column in the revised manuscript:

Line 158-159: "The particulate iSOA OSs were analyzed by a LC-MS employing a reversed-phase column ($C_{18}$, 2.1 mm ×100 mm, 1.7 μm, Waters)."

2.) Related to # 1, can you provide at least 1 example TIC from both GC/MS and LC/MS analyses of the targeted iSOA tracers in your field samples in the Supporting Information (SI) file? I suspect reviewers will want to see that you can carefully resolve these compounds from each other, especially if you used RPLC to measure isoprene-derived OSs.

**Response:** Thanks for the editor's suggestion. We have added a figure in SI showing the examples of TIC of iSOA tracers (and also EIC of OSs) in ambient $PM_{2.5}$ samples from both GC-MS and LC-MS analyses.

[Figure]

Figure S1. Example chromatograms of iSOA tracers in ambient $PM_{2.5}$ samples. (a) Total ion chromatogram (TIC) of iSOA OSs from the LC-MS analysis; (b) and (c) Extracted ion chromatograms of 2-methyltetrol sulfate (2-MT-OS) and 2-methylglyceric acid sulfate (2-MG-OS), respectively; (d) TIC of a trimethylsilyl-derivatized iSOA polyol tracers from the GC-MS analysis.

3.) Related to # 2, if you did in fact use RPLC, you will need to discuss how this adds a large degree of uncertainty in your quantitative results (beyond the lack of authentic compounds for LC/MS). Since isoprene-derived OSs typically all co-elute on an RPLC column (with other water-soluble compounds like inorganics and organics), you may underestimate some of these due to ion suppression and matrix effects strongly associated with electrospray ionization (ESI). This is why Betsy Stone's group and my

group developed HILIC instead of RPLC to chromatographically resolve and quantify isoprene-derived OSs before ESI and mass spec detection.

**Response:** Thanks for the editor's comment. Both reviewers also raised this concern. In the present work, because of lack of authentic standards of isoprene-derived OSs, we are not able to quantify the absolute value of underestimation in the concentration of 2-MT-OS and 2-MG-OS due to the matrix effect. However, using ambient $PM_{2.5}$ samples with different concentrations, we can quantify the relative extent of underestimation in OS concentrations due to matrix effect in different samples, which allows for an evaluation of uncertainties in the abundance, trend, and relative ratios of different isoprene SOA (iSOA) tracers and the validity of the conclusions in this study.

To this end, we have conducted a set of experiments where the extracts of ambient $PM_{2.5}$ samples with different concentrations were mixed and the measured signals of 2-MT-OS and 2-MG-OS in mixed extracts were compared to the sum of OS signals detected separately in individual extracts. The concentrations of $PM_{2.5}$, sulfate, as well as 2-MT-OS and 2-MG-OS in ambient samples used for this evaluation are listed in Table S3. The relative matrix effect factor ($F_{matrix}$), defined as the ratio of the measured OS signals in mixed extracts to the sum of OS signals measured in each extract before mixing, are used to evaluate the matrix effects of OSs. A $F_{matrix}$ value of less than 1 indicates the presence of matrix effect.

As shown in Figure S2, the $F_{matrix}$ values were significantly smaller than 1 in both summer and winter, indicating that the signal responses of 2-MT-OS and 2-MG-OS in mixed extracts were largely suppressed due to the matrix effect. Notably, $F_{matrix}$ exhibited a significant negative dependence on the reduced mass ($\mu$, $\mu g\ m^{-3}$), a proxy used to represent effective mass loadings of the mixed $PM_{2.5}$ extracts, defined as:

$$\mu = \sqrt{m_1 * m_2 / (m_1 + m_2)} \tag{1}$$

where $m_1$ and $m_2$ are the $PM_{2.5}$ mass loading of individual samples. This observation suggests that the concentrations of iSOA OSs in $PM_{2.5}$ samples collected in 2015 were underestimated more significantly than those in 2021, given that ambient $PM_{2.5}$ concentrations declined by 39.8% and 47.0% from 2015 to 2021 in summer and winter, respectively.

As $PM_{2.5}$ concentrations in ambient samples used for the matrix effect evaluation generally represent lower (e.g., Low 1 and Low 3 in Table S3) or upper (e.g., High 1 and High 3) ends of $PM_{2.5}$ concentrations during the observation period, the relative differences in $F_{matrix}$ values measured at varying reduced mass (Figure S2) may roughly reflect the differences in the extent of underestimation in OS concentrations for samples collected across 2015-2021. During summer, the $F_{matrix}$ value decreased from 0.71 to 0.63 for 2-MT-OS and from 0.85 to 0.58 for 2-MG-OS with increasing reduced mass, indicating that the concentrations of these two iSOA OSs were a factor of 1.2 and 1.5 more underestimated in 2015 than in 2021 due to matrix effect. Similarly, during winter the $F_{matrix}$ values of iSOA OSs decreased from 0.9 to 0.6 with rising reduced mass, implying a factor of 1.5 greater underestimation in OS concentrations in 2015 than in 2021. In this study, the measured concentration of 2-MT-OS exhibited a decreasing inter-annual trend, while that of 2-MG-OS did not vary significantly between 2015-2021. Accounting for the significantly larger matrix effects in 2015 samples compared to 2021 samples, the true concentrations of 2-MT-OS would show a sharper decrease than observed here and 2-MG-OS would exhibit a declining trend.

The matrix effect of iSOA OSs was also considered when discussing their relative ratios and correlation analysis. Overall, after accounting for the underestimation of OSs, the abundance of iSOA OSs would dominate over that of polyol tracers, consistent with previous field observations in urban areas employing HILIC-MS. In addition, the uncertainty due to matrix effect does not change the correlation analysis results and the conclusion regarding the driving factors for iSOA formation. The detailed revisions made to the manuscript have been provided in our responses to Reviewer #1's comments.

Table S3. Major components in eight PM$_{2.5}$ filter samples that used for estimating matrix effect of 2-MT-OS and 2-MG-OS.

| Season | Sample | Sampling date | PM$_{2.5}$ (µg m$^{-3}$) | SO$_4^{2-}$ (µg m$^{-3}$) | 2-MG-OS (ng m$^{-3}$) | 2-MT-OS (ng m$^{-3}$) |
|--------|--------|--------------|----------|----------|---------|---------|
| Summer | High 1 | 2015/7/28 | 35.66 | 7.74 | 13.04 | 157.89 |
|        | High 2 | 2019/7/24 | 31.76 | 4.58 | 12.36 | 100.61 |
|        | Low 1  | 2019/8/5  | 10.04 | 2.29 | 0.58 | 0.83 |
|        | Low 2  | 2021/8/5  | 12.21 | 1.48 | 0.18 | 0.18 |
| Winter | High 3 | 2022/1/2  | 77.76 | 6.28 | 1.80 | 0.47 |
|        | High 4 | 2022/1/3  | 65.38 | 5.41 | 1.54 | 0.41 |
|        | Low 3  | 2021/12/25 | 18.04 | 3.42 | 0.49 | 0.07 |
|        | Low 4  | 2021/12/26 | 23.71 | 3.01 | 0.70 | 0.08 |

[Figure]

Figure S2. Matrix effect factors (F$_{matrix}$, defined as the ratio of signal response in mixed extracts to the sum of signal response in individual extracts) of 2-MT-OS and 2-MG-OS under different mixture type in summer (a) and winter (b) (yellow background represents low-plus-low mixture, green background represents high-plus-low mixture, and blue background represents high-plus-high mixture).

4.) With regards to Table 1, you should be aware of a paper by Chen et al. (2024, ACS ES&T Air) that simulated time-resolved iSOA components (2-methyltetrols and methyltetrol sulfates). From these simulations they were able to recommend kinetic values: https://pubs.acs.org/doi/abs/10.1021/acsestair.4c00002

**Response:** We appreciate the editor's comment. We have incorporated the kinetic values recommended by Chen et al. (2024) (referred to as $k_{i,j\text{-IEPOX}}^{-5}$, see Table 1) into our estimates of aerosol reactivity toward IEPOX. The reactive uptake coefficients of IEPOX estimated using $k_{i,j\text{-IEPOX}}^{-5}$ were close to that estimated with $k_{i,j\text{-IEPOX}}^{-4}$. However, the $k_{i,j\text{-IEPOX}}^{-5}$ did not provide kinetic values ($k_{i,\text{HSO4}^-}$) for the third-order reaction

rate constants of IEPOX with nucleophiles catalyzed by bisulfate, and a sensitivity test suggests that these bisulfate-catalyzed reactions played a role in controlling the trend of aerosol reactivity ($k_{het}$). As a result, we used $k_{i,j\text{-IEPOX}}^{-4}$ to predict aerosol heterogeneous reactivity toward IEPOX.

We have included the analysis with $k_{i,j\text{-IEPOX}}^{-5}$ and updated relevant discussions in the revised manuscript.

L532-534: "More recently, …Chen et al. (2024) constrained reaction rate constant of IEPOX ($k_{i,j\text{-IEPOX}}^{-5}$) using a phase-separation box model with chamber measurements."

L554-566: "Figure 5 shows the γ and $k_{het}$ values for IEPOX and HMML&MAE estimated using different sets of kinetic parameters listed in Table 1. The CMAQ-modeled values are also displayed for comparison. The $k_{het}$ estimated by $k_{i,j\text{-IEPOX}}^{-1}$, $k_{i,j\text{-IEPOX}}^{-2}$ and $k_{i,j\text{-IEPOX}}^{-5}$ in summer had highest values in 2019. This might be attributed to the fact that these three sets of parameters lack the third-order reaction rate constant of IEPOX with nucleophiles catalyzed by bisulfate (Table 1). As a comparison, we calculated the $k_{het}$ of IEPOX and HMML&MAE excluding the reaction rate constant catalyzed by bisulfate using $k_{i,j\text{-IEPOX}}^{-3}$ and $k_{i,j\text{-IEPOX}}^{-4}$ (Figure S10). We found that the inter-annual trend of $k_{het}$ for IEPOX (Figure S10a) and HMML&MAE (Figure S10b) in summer was altered and similar to that of IEPOX calculated by $k_{i,j\text{-IEPOX}}^{-1}$, $k_{i,j\text{-IEPOX}}^{-2}$ and $k_{i,j\text{-IEPOX}}^{-5}$, while the inter-annual trend of $k_{het}$ in winter was not sensitive to the exclusion of reaction rate constant catalyzed by bisulfate. This result indicates a contribution of nucleophilic-addition of epoxides catalyzed by bisulfate to the heterogeneous reactivity. It also suggests that the $k_{i,j\text{-IEPOX}}^{-4}$ is more appropriate for predicting aerosol heterogeneous reactivity toward IEPOX than other four sets of kinetic parameters"

Table 1. Third-order reaction rate constants of IEPOX and HMML&MAE with sulfate and water in the aqueous phase

| | $k_{i,H+}$ ($M^{-2}\cdot s^{-1}$) | | $k_{i,HSO4^-}$ ($M^{-2}\cdot s^{-1}$) | | References |
|---|---|---|---|---|---|
| | i=$H_2O$ | i=$SO_4^{2-}$ | i=$H_2O$ | i=$SO_4^{2-}$ | |
| $k_{i,j\text{-IEPOX}}^{-1}$ | $5.3\times10^{-2}$ | $5.2\times10^{-1}$ | — | — | (Piletic et al., 2013) |
| $k_{i,j\text{-IEPOX}}^{-2}$ | $3.4\times10^{-4}$ | $4.8\times10^{-4}$ | — | — | (Riedel et al., 2015) |
| $k_{i,j\text{-IEPOX}}^{-3}$ | $9\times10^{-4}$ | $2\times10^{-4}$ | $1.3\times10^{-5}$ | $2.9\times10^{-6}$ | (Pye et al., 2013) |
| $k_{i,j\text{-IEPOX}}^{-4}$ | $9\times10^{-4}$ | $8.8\times10^{-3}$ | $1.3\times10^{-5}$ | $2.9\times10^{-6}$ | (Pye et al., 2017) |
| $k_{i,j\text{-IEPOX}}^{-5}$ | $5.3\times10^{-4}$ | $5.2\times10^{-3}$ | — | — | (Chen et al., 2024) |
| $k_{i,j\text{-HMML\&MAE}}$ | $9\times10^{-4}$ | $2\times10^{-4}$ | $1.3\times10^{-5}$ | $2.9\times10^{-6}$ | (Pye et al., 2013) |

[Figure]

Figure 5. Reactive uptake coefficients (γ) and pseudo-first-order heterogeneous reaction rate constant

($k_{het}$, s$^{-1}$) of gas-phase IEPOX and HMML&MAE estimated using different sets of $k_{i,j\text{-IEPOX}}$ and $k_{i,j\text{-HMML\&MAE}}$ listed in Table 1 (a and c) and simulated by CMAQ model in Case 1 (b and d).

5.) Lastly, there has been recent work published showing that iSOA can be heterogeneously oxidized by OH radicals (providing an aging mechanism), yielding products (many are OS products) that can be measured by LC/MS techniques (especially HILIC methods). This includes lab studies by Chen et al. (2020, ES&T Letters, https://doi.org/10.1021/acs.estlett.0c00276), Yan et al. (2023, ACS Earth and Space Chemistry, https://doi.org/10.1021/acsearthspacechem.3c00073) and Yan et al. (2025, J. Phys. Chem. A, https://pubs.acs.org/doi/10.1021/acs.jpca.4c08082), and these lab studies confirmed earlier findings of heterogeneous OH oxidation products of iSOA in ambient aerosol by Hettiyadura et al. (2019, https://doi.org/10.5194/acp-19-3191-2019) from Atlanta, Georgia.

Furthermore, Frauenheim et al. (2024, ES&T, https://doi.org/10.1021/acs.est.4c06850) demonstrated that the "C5-alkene triols", or more specifically the 3-methylenebutane-1,2,4-triol and 3-methyltetrahydrofuran-2,4-diols (which are both considered in-particle isomerization products of IEPOX in wet acidic aerosols) can off gas from aerosols and then be oxidized in the air by OH radical to produce other types of iSOA constituents. Did you see any of these in your analysis?

**Response:** Yes, we have observed several OS species, including $C_2H_3O_5S^-$, $C_2H_3O_6S^-$, $C_3H_5O_6S^-$, $C_4H_7O_7S^-$ (an isomer of 2-MG-OS), $C_5H_7O_7S^-$, and $C_5H_9O_7S^-$, which can be formed through the heterogeneous oxidation of 2-MT-OS by OH radicals, as proposed by Chen et al. (2020). However, in the present work, we did not examine the formation and abundance of these aging products and instead focus on the iSOA tracers. We considered the removal of iSOA polyol tracers by OH radicals in both gas and particle phases in the CMAQ model simulations (see Section 2.5 in the main text). The results demonstrate that the CMAQ model reasonable well predicted the concentrations of iSOA tracers and successfully captured the declining inter-annual trends of total iSOA, 2-MT-OS, and 2-MG-OS in both summer and winter.

In this study, the gas-phase concentrations of 3-methylenebutane-1,2,4-triol (3-MBT) and 3-methyltetrahydrofuran-2,4-diols were not measured, thus it is difficult to quantitatively evaluate the contribution of their oxidation products to other types of iSOA constituents such as 2-MT-OS and 2-MG-OS. However, given that concentrations of IEPOX-SOA observed here showed a significant decreasing trend over the observation period, the concentrations of IEPOX isomerization products, such as $C_5$-alkene triols and 3-methyltetrahydrofuran-2,4-diols, likely also decreased annually. Therefore, gas-phase OH oxidation of these species might contribute to the inter-annual decline in 2-MT-OS and 2-MG-OS.

We have added a discussion regarding the potential influence of gas-phase OH oxidation of $C_5$-alkene triols and 3-methyltetrahydrofuran-2,4-diols on the trend of 2-MT-OS and 2-MG-OS in the revised manuscript.

L378-387: "Recently, Frauenheim et al. (2024) demonstrated that gas-phase OH oxidation of 3-methylenebutane-1,2,4-triol (3-MBT) and 3-methyltetrahydrofuran-2,4-diols, formed from acid-catalyzed isomerization of IEPOX in aerosols, can yield 2-MT-OS and 2-MG-OS. However, in the present study, the gas-phase concentrations of 3-MBT and 3-methyltetrahydrofuran-2,4-diols were not measured, precluding a quantitative assessment of the contribution of their oxidation products to 2-MT-OS and 2-MG-OS. Given that IEPOX-SOA concentrations observed here exhibited a significant decreasing trend over the observation period, the concentrations of IEPOX isomerization products, such as 3-MBT and 3-methyltetrahydrofuran-2,4-diols, likely also decreased annually. Therefore, gas-phase OH oxidation of these species might represent a plausible contributor to the inter-annual decline in 2-MT-OS and 2-MG-OS."

We are grateful to the reviewer for the thoughtful comments, which are very helpful for improving our manuscript. Our point-to-point responses to each comment are as follows (the reviewer's comments are in black text, our responses are in blue text, and revised texts that appear in the manuscript are in red text).

General Comments:

This study presents measurements of isoprene and its SOA products in Shanghai across summer and wintertime in 2015, 2019, 2021. The study examines isoprene SOA response to emissions reductions, particularly the hydroperoxy pathway (IEPOX-SOA) and NOx-dominated pathway (HMML, MAE). CMAQ modeling captures some trends in experimental data, including a decreasing IEPOX-SOA over the study period. The study concludes that emissions reductions can decrease biogenic SOA in urban areas, and that a deeper understanding of isoprene-derived SOA is needed.

I have major technical concerns about the manuscript that should be addressed prior to reconsideration for publication. Additionally, there are numerous opportunities for more thorough comparisons of the modeled and experimental data. Similarly, there is opportunity for thorough and meaningful comparisons of the current study to literature more specific to urban areas.

Response: Thanks for the reviewer's comments. To address the reviewer's technical concerns, we have conducted additional experiments to evaluate the matrix effect of iSOA OSs. The evaluation shows that the matrix effect of OSs exhibited a strong dependence on the $PM_{2.5}$ mass concentration. Accounting for the significantly larger matrix effects for $PM_{2.5}$ samples collected in 2015 than in 2021, the abundance of 2-MT-OS would exhibit a sharper decreasing tend and 2-MG-OS would also show a declining trend. Overall, these observations are consistent with CMAQ model simulations, which predicted a decreasing inter-annual trend for both 2-MT-OS and 2-MG-OS during the observation period.

In the revised manuscript, we have added additional analyses and discussions about the matrix effect on the abundance, inter-annual trend, and relative ratios between different pathway products. Additionally, we have provided more thorough comparisons between modeled and measured data, as well as between the current study and literature (see our responses to specific comments below).

Specific Comments:

1. I have major technical concerns about the potential for matrix effects and interferences in the selected analytical measurements, and how these may influence the results of the current study.

a.    Recent work demonstrates that use of reversed phase liquid chromatography (RPLC) with mass spectrometry (MS) can caused significant underestimation of some isoprene-derived organosulfates (OS) with little retention on reversed phase LC columns (Liang et al. 2025, pubs.acs.org/doi/10.1021/acs.est.5c01846). OS co-elute from the RPLC column with sulfate and other inorganic ions, which can suppress OS signal. Changes to the aerosol matrix can thus be misinterpreted as changes in OS concentrations. Because the author's OS measurement method is likely subject to matrix suppression, there is concern about the validity of the OS concentrations and relative ratios presented in this study.

Response: Thanks for the reviewer's comment. Previous studies have illustrated that the use of PRLC-MS in the quantification of low-molecular-weight OS compounds could cause significant matrix effect (Liang et al., 2025; Liu et al., 2024; Bryant et al., 2020). In the present work, because of lack of authentic standards of isoprene-derived OSs, we are not able to quantify the absolute value of underestimation in the concentration of 2-MT-OS and 2-MG-OS due to the matrix effect. However, using ambient $PM_{2.5}$ samples with different concentrations, we can quantify the relative extent of underestimation in OS concentrations due to matrix effect in different samples, which allows for an evaluation of uncertainties in the abundance, trend, and relative ratios of different isoprene SOA (iSOA) tracers and the validity of the major conclusions in this study.

To this end, we have conducted a set of experiments where the extracts of ambient $PM_{2.5}$ samples with different concentrations were mixed and the measured signals of 2-MT-OS and 2-MG-OS in mixed extracts were compared to the sum of OS signals detected separately in individual extracts. The concentrations of $PM_{2.5}$, sulfate, as well as 2-MT-OS and 2-MG-OS in ambient samples used for this evaluation are listed in Table S3. The relative matrix effect factor ($F_{matrix}$), defined as the ratio of the

measured OS signals in mixed extracts to the sum of OS signals measured in each extract before mixing, are used to evaluate the matrix effects of OSs. A $F_{matrix}$ value of less than 1 indicates the presence of matrix effect.

As shown in Figure S2, the $F_{matrix}$ values were significantly smaller than 1 in both summer and winter, indicating that the signal responses of 2-MT-OS and 2-MG-OS in mixed extracts were largely suppressed due to the matrix effect. Notably, $F_{matrix}$ exhibited a significant negative dependence on the reduced mass ($\mu$, $\mu g\ m^{-3}$), a proxy used to represent effective mass loadings of the mixed $PM_{2.5}$ extracts, defined as:

$$\mu = \sqrt{m_1 * m_2/(m_1 + m_2)}$$

where $m_1$ and $m_2$ are the $PM_{2.5}$ mass loading of individual samples. This observation suggests that the concentrations of iSOA OSs in $PM_{2.5}$ samples collected in 2015 were underestimated more than those in 2021, given that ambient $PM_{2.5}$ concentrations declined by 39.8% and 47.0% from 2015 to 2021 in summer and winter, respectively.

As $PM_{2.5}$ concentrations in ambient samples used for the matrix effect evaluation generally represent lower (e.g., Low 1 and Low 3 in Table S3) or upper (e.g., High 1 and High 3) ends of $PM_{2.5}$ concentrations during the observation period, the relative differences in measured $F_{matrix}$ values at varying reduced mass (Figure S2) may roughly reflect the differences in the extent of underestimation in OS concentrations for samples collected across 2015-2021. During summer, the $F_{matrix}$ value decreased from 0.71 to 0.63 for 2-MT-OS and from 0.85 to 0.58 for 2-MG-OS with increasing reduced mass, indicating that the concentrations of these two iSOA OSs were a factor of 1.2 and 1.5 more underestimated in 2015 than in 2021 due to matrix effect. Similarly, during winter the $F_{matrix}$ values of iSOA OSs decreased from 0.9 to 0.6 with rising reduced mass, implying a factor of 1.5 greater underestimation in OS concentrations in 2015 than in 2021. In this study, the measured concentration of 2-MT-OS exhibited a decreasing inter-annual trend, while 2-MG-OS showed insignificant variation between 2015-2021. Accounting for the significantly larger matrix effects in 2015 samples compared to 2021 samples, the true concentrations of 2-MT-OS would show a sharper decrease and 2-MG-OS would exhibit a declining trend.

Table S3. Major components in eight $PM_{2.5}$ filter samples used for evaluating matrix effect of 2-MT-OS and 2-MG-OS during LC-MS analysis.

| Season | Sample | Sampling date | $PM_{2.5}$ ($\mu g\ m^{-3}$) | $SO_4^{2-}$ ($\mu g\ m^{-3}$) | 2-MG-OS ($ng\ m^{-3}$) | 2-MT-OS ($ng\ m^{-3}$) |
|--------|--------|---------------|------------|-----------|---------|---------|
| Summer | High 1 | 2015/7/28 | 35.66 | 7.74 | 13.04 | 157.89 |
|        | High 2 | 2019/7/24 | 31.76 | 4.58 | 12.36 | 100.61 |
|        | Low 1  | 2019/8/5  | 10.04 | 2.29 | 0.58 | 0.83 |
|        | Low 2  | 2021/8/5  | 12.21 | 1.48 | 0.18 | 0.18 |
| Winter | High 3 | 2022/1/2  | 77.76 | 6.28 | 1.80 | 0.47 |
|        | High 4 | 2022/1/3  | 65.38 | 5.41 | 1.54 | 0.41 |
|        | Low 3  | 2021/12/25 | 18.04 | 3.42 | 0.49 | 0.07 |
|        | Low 4  | 2021/12/26 | 23.71 | 3.01 | 0.70 | 0.08 |

[Figure]

Figure S2. Relative matrix effect factors ($F_{matrix}$, defined as the ratio of signal response in mixed extracts to the sum of signal response in individual extracts) of 2-MT-OS and 2-MG-OS under different mixture type in summer (a) and winter (b) (yellow background represents low-plus-low mixture, green background represents high-plus-low mixture, and blue background represents high-plus-high mixture).

The significant matrix effect of iSOA OSs was also considered when discussing their relative ratios of different tracers. Considering the stronger matrix effects for 2015 samples than for 2019 and 2021 samples, the downward trend of 2-MT-OS/2-MTs would be sharper, while the slight upward trend of 2-MG-OS/2-MG might be reversed since the true trend of 2-MG-OS was downward and 2-MG had no significant variation.

We have revised the associated content in the manuscript as follows:

Lines 221-255: "Previous studies have demonstrated that the concentrations of 2-MT-OS and 2-MG-OS were significantly underestimated due to matrix effect by using reversed phase liquid chromatography-mass spectrometry (RPLC-MS) (Hettiyadura et al., 2015; Bryant et al., 2020; Bryant et al., 2021). In the present work, because of lack of authentic standards of isoprene-derived OSs, we are not able to quantify the absolute value of underestimation in the concentration of 2-MT-OS and 2-MG-OS due to the matrix effect. However, using ambient $PM_{2.5}$ samples with different concentrations, we can quantify the relative extent of underestimation in OS concentrations due to matrix effect in different samples, which allows for an evaluation of uncertainties in the abundance, trend, and relative ratios of different iSOA tracers in this study. In the matrix effect experiments, the extracts of ambient $PM_{2.5}$ samples with different concentrations were mixed and the measured signals of 2-MT-OS and 2-MG-OS in mixed extracts were compared to the sum of OS signals detected separately in individual extracts. The concentrations of $PM_{2.5}$, sulfate, as well as 2-MT-OS and 2-MG-OS in ambient samples used for this evaluation are listed in Table S3. The relative matrix effect factor ($F_{matrix}$), defined as the ratio of the measured OS signals in mixed extracts to the sum of OS signals measured in each extract before mixing, are used to evaluate the matrix effect of OSs. A $F_{matrix}$ value of less than 1 indicates the presence of matrix effect.

As shown in Figure S2, the $F_{matrix}$ values were significantly smaller than 1 in both summer and winter, indicating that the signal responses of 2-MT-OS and 2-MG-OS in mixed extracts were largely suppressed due to the matrix effect. Notably, $F_{matrix}$ exhibits a significant negative dependence on the reduced mass ($\mu$, $\mu g\ m^{-3}$), a proxy used to represent effective mass loadings of the mixed $PM_{2.5}$ extracts, defined as:

$$\mu = \sqrt{m_1 * m_2 / (m_1 + m_2)} \qquad (1)$$

where $m_1$ and $m_2$ are the $PM_{2.5}$ mass loading of individual samples. This observation suggests that the concentrations of iSOA OSs in $PM_{2.5}$ samples collected in 2015 were underestimated more than those in

2021, given that ambient $PM_{2.5}$ concentrations declined by 39.8% and 47.0% from 2015 to 2021 in summer and winter, respectively.

As $PM_{2.5}$ concentrations in ambient samples used for the matrix effect evaluation generally represent lower or upper ends of $PM_{2.5}$ concentrations during the observation period (see Table S3), the relative differences in measured $F_{matrix}$ values at varying reduced mass (Figure S2) may roughly reflect the differences in the extent of underestimation in OS concentrations for samples collected across 2015-2021. During summer, the $F_{matrix}$ value decreased from 0.71 to 0.63 for 2-MT-OS and from 0.85 to 0.58 for 2-MG-OS with increasing reduced mass, indicating that the concentrations of these two iSOA OSs were a factor of 1.2 and 1.5 more underestimated in 2015 than in 2021 due to matrix effect. Similarly, during winter the $F_{matrix}$ values of iSOA OSs decreased from 0.9 to 0.6 with rising reduced mass, implying a factor of 1.5 greater underestimation in OS concentrations in 2015 than in 2021."

Lines 371-379: "However, the inter-annual trend of iSOA OSs could be altered due to the matrix effect. The measured concentration of 2-MT-OS exhibited a decreasing inter-annual trend, while 2-MG-OS showed insignificant variation between 2015-2021. Accounting for the significantly larger matrix effects in 2015 samples compared to 2021 samples (see Section 2.3), the true concentrations of 2-MT-OS would decrease more sharply and 2-MG-OS might also exhibit a declining trend."

b.    There are additional concerns about $C_5$ alkene-triols being artifacts of gas chromatography (GC) MS analysis (Frauenheim, et al. doi/10.1021/acs.estlett.2c00548). The extent to which these may be artifacts in the current study should be considered, especially following the result that they are the dominant product observed by GCMS.

Response: Thanks for the reviewer's comment. Frauenheim et al. (2022) have found that the thermal decomposition of 3-methyltetrahydrofuran-2,4-diols (less than 15%) could transfer to two isomers of $C_5$-alkene trols (cis-/trans-3-methyl-but-3-ene-1,2,4-triols) during GC/MS analysis. In our work, we quantified the concentrations of 3-methyltetrahydrofuran-2,4-diols using 2-methylerythritol as a surrogate standard. The concentrations of 3-methyltetrahydrofuran-2,4-diols were less than 5% of $C_5$-alkene triols in summer but had comparable levels in winter. This result indicates that 3-methyltetrahydrofuran-2,4-diols was a minor contributor to $C_5$-alkene triols in summer but an important source for $C_5$-alkene triols in winter.

In addition, the thermal degradation of 2-MT-OS and its oligomers during GC-MS analysis could be a potential contributor to $C_5$-alkene triols. Cui et al. (2018) found that thermal degradation of 2-MT-OS could generate all three isomers of $C_5$-alkene triols and such processes accounted for 14.7 and 42.7% of $C_5$-alkene triols observed in urban Manaus, Brazil and southeastern U.S., respectively. In contrast, Yee et al. (2020) found that the thermal decomposition of 2-MT-OS could only produce one isomer, 3-methyl-2,3,4-trihydroxy-1-butene. In this study, we are not able to conduct a quantitative assessment of the transformation of 2-MT-OS to $C_5$-alkene triols during GC-MS analysis due to the lack of authentic standards. Assuming that all the isomers of $C_5$-alkene triols could come from the thermal degradation of 2-MT-OS, a considerable fraction of $C_5$-alkene triols could be artifacts since the concentrations of 2-MT-OS (with matrix effect considered) were significantly higher than $C_5$-alkene triols. However, if only 3-methyl-2,3,4-trihydroxy-1-butene was the product of 2-MT-OS degradation, the concentrations of $C_5$-alkene triols would be overestimated by no more than 23.8% since the 3-methyl-2,3,4-trihydroxy-1-butene on average accounted for 23.8% of the concentrations of $C_5$-alkene triols. Thus, $C_5$-alkene triols were likely overestimated mainly due to the thermal decomposition of 2-MT-OS, with the 3-methyltetrahydrofuran-2,4-diols likely being an important source in winter.

Because of the lack of authentic standards of $C_5$-alkene triols, it is challenging to quantify artifacts resulting from the thermal degradation of 2-MT-OS during GC/MS analysis in the present work. In addition, inconsistent results regarding the quantification uncertainties of $C_5$-alkene triols have been reported in the literature (Cui et al., 2018; Frauenheim et al., 2022). Therefore, we do not discuss the abundance or inter-annual trends of $C_5$-alkene triols in detail in this study, but instead focus on 2-MTs, 2-MG, and OSs. In addition, they were excluded from the correlation analysis. When discussing the relative ratios (including IEPOX-SOA/HMMML&MAE-SOA and iSOA OSs/polyol tracers), the measurement uncertainties of $C_5$-alkene triols were considered.

The manuscript has been revised as follows:

Lines 354-371: "However, the concentrations of $C_5$-alkene triols might be overestimated since previous studies have reported that concentrations of $C_5$-alkene triols could be artifacts of thermal degradation products of 3-methyltetrahydrofuran-2,4-diols and 2-MT-OS during GC/MS analysis (Cui et al., 2018;

Frauenheim et al., 2022). Frauenheim et al. (2022) found that less than 15% of 3-methyltetrahydrofuran-2,4-diols could transfer to two isomers of $C_5$-alkene triols (cis-/trans-3-methyl-but-3-ene-1,2,4-triols). In the present study, using 2-methylerythritol as a surrogate standard, the concentrations of 3-methyltetrahydrofuran-2,4-diols were determined to be less than 5% of $C_5$-alkene triols in summer but had comparable concentrations to $C_5$-alkene triols in winter. This result indicates that 3-methyltetrahydrofuran-2,4-diols was a minor contributor to $C_5$-alkene triols in summer but an important source for $C_5$-alkene triols in winter. In contrast, the contribution from the thermal degradation of 2-MT-OS might be more significant, though the specific contribution remains to be quantified; Cui et al. (2018) found that thermal degradation of 2-MT-OS could generate all three isomers of $C_5$-alkene triols, while Yee et al. (2020) found that the thermal decomposition of 2-MT-OS could only produce one isomer, 3-methyl-2,3,4- trihydroxy-1-butene. Given these uncertainties, it is difficult to quantitatively evaluate the artifact formation of $C_5$-alkene triols during GC/MS analysis. Therefore, the abundance and inter-annual trend of $C_5$-alkene triols were not discussed in detail in the present work."

Lines 398-404: "Although $C_5$-alkene triols might be largely artifacts of GC/MS analysis (Cui et al., 2018; Frauenheim et al., 2022), the concentrations of IEPOX-SOA excluding $C_5$-alkene triols were still dominant over HMML&MAE-SOA in summer. In addition, previous studies have demonstrated that the concentrations of 2-MT-OS were underestimated more than 2-MG-OS by a factor of 5.7-9.1 in Beijing (Bryant et al., 2020) and 2.9 in Guangzhou (Bryant et al., 2021). If 2-MT-OS was also more significantly underestimated than 2-MG-OS in the present study, the predominance of IEPOX-SOA over HMML&MAE-SOA would be more pronounced."

Lines 411-413: "However, given the significant underestimation of iSOA OSs due to matrix effect and overestimation of $C_5$-alkene triols due to their potential artifact formation, the true concentration of iSOA OSs would predominate over that of polyol tracers."

c.    Taken together, the suppression of OS and potential artifact formation of $C_5$ alkene triols raises questions to the validity of the paragraph that discusses "the dominance of iSOA polyol tracers over OS tracers…"   With one signal being enhanced and the other suppressed, such comparisons have very large uncertainties.

Response: Thanks for the reviewer's suggestion. We have added a discussion about the uncertainties from the suppression of OS and potential artifact formation of $C_5$-alkene triols in manuscript. Considering both the underestimation of iSOA OSs due to matrix effect and overestimation of $C_5$-alkene triols (as discussed in response to comment #1b), iSOA OSs were found to predominate over polyol tracers. The text in manuscript was revised as follows:

Lines 411-418: "However, given the significant underestimation of iSOA OSs due to matrix effect and overestimation of $C_5$-alkene triols due to their potential artifact formation, the true concentration of iSOA OSs would predominate over that of polyol tracers. The iSOA OSs prevailing over polyol tracers is consistent with urban observations using HILIC-MS, such as in Manaus, Brazil (Cui et al., 2018) and Guangzhou, China (Liu et al., 2025) (see Table S6). Using RPLC-MS, Bryant et al. (2020) also observed higher concentrations of iSOA OSs than polyol tracers in Beijing, China. Considering the potential underestimation of iSOA OSs due to matrix effect, the concentration of iSOA OSs would be even higher than that of polyol tracers in their study."

d.    The potential for bias in measurements mentioned should be thoroughly considered and discussed in downstream calculations and comparisons, including SOA estimates, relative ratios of isoprene SOA products, correlation analysis, model comparisons, etc.

Response: Thanks for the reviewer's comment. We have added detailed analyses and discussions regarding the uncertainties in the measurements of OSs and $C_5$-alkene triols, as well as the subsequent SOA estimates and relative ratios of isoprene SOA products (see our responses above). We have also modified the discussions about comparisons between measurements and model simulations of iSOA tracers (see below).

Lines 439-446: "For iSOA tracers, the Case 1 showed a better prediction than the Base Case. Overall, the simulated IEPOX-SOA tracers were biased low in summer, but biased high in winter (Figure 3b and 3c). In contrast, the 2-MG and 2-MG-OS were biased low in both seasons (Figure 3d and 3e). The underestimation of 2-MG is consistent with previous simulations at 14 sites across China in the summer of 2012 (Qin et al., 2018). Accounting for the underestimation of OSs due to the matrix effect, simulated concentrations of 2-MT-OS would be more biased low in summer but might be close to observations in winter. Similarly, the under-prediction of 2-MG-OS would be more significant in both seasons."

In addition, we have conducted a sensitivity test to evaluate the influence of measurement uncertainties on the correlation analysis between iSOA tracers and different influencing factors. Recently, Liang et al. (2025) has illustrated that the concentrations of low-molecular-weight ($C_2$ and $C_3$) OSs quantified with RPLC-MS were 1-2 orders of magnitudes lower than those measured with HILIC-MS duo to the matrix effect. Additionally, both the results in Liang et al. (2025) and the matrix effect experiments in our work have found a greater signal suppression of OSs at higher $PM_{2.5}$ mass loadings. Since the retention time of 2-MT-OS and 2-MG-OS was very close to $C_{2-3}$ OSs (see Figure S1a), we roughly assumed that the concentrations of 2-MT-OS and 2-MG-OS were also underestimated by up to 100 times and that the underestimation extent is linearly dependent on the $PM_{2.5}$ concentration. We then performed a correlation analysis using the corrected concentrations of 2-MT-OS and 2-MG-OS. As shown in Tables S4, iSOA OSs still exhibited the strongest correlation with the concentrations of $O_3$ and Ox in summer and sulfate, nitrate, and LWC in winter, although the correlation coefficients ($r^2$) with $O_3$ and $O_x$ were slightly decreased (less than 0.1) while those with sulfate increased by 0.1-0.4 compared to the correlation analysis with the observed data. This result indicated that the measurement uncertainties did not significantly influence the correlation analysis and the evaluation of the dominant influencing factors for the formation of iSOA OSs in this study.

We have added a statement regarding the sensitivity test result to the main text and the details about this test to Section S5 in the supplement:

Lines 499-500: "A sensitivity test considering the measurement uncertainties of iSOA tracers did not significantly influence the correlation analysis results (see details in Section S5)."

Lines 103-121 in the supplement:

**"S5. Evaluation of the influence of matrix effect on correlation analysis of iSOA OSs**

We have conducted a sensitivity test to evaluate the influence of measurement uncertainties on the correlation analysis between iSOA OSs and different influencing factors. Recently, Liang et al. (2025) has illustrated that the concentrations of low-molecular-weight ($C_2$ and $C_3$) OSs quantified with RPLC-MS were 1-2 orders of magnitudes lower than those measured with HILIC-MS duo to the matrix effect. Additionally, both the results in Liang et al. (2025) and the matrix effect experiments in our work have found a greater signal suppression of OSs at higher $PM_{2.5}$ mass loadings. Since the retention time of 2-MT-OS and 2-MG-OS was very close to $C_{2-3}$ OSs (see Figure S1a), we roughly assumed that the concentrations of 2-MT-OS and 2-MG-OS were also underestimated by up to 100 times and that the underestimation extent was linearly dependent on the $PM_{2.5}$ concentration. We then performed a correlation analysis using the modified concentrations of 2-MT-OS and 2-MG-OS. As shown in Tables S7, iSOA OSs still exhibited the strongest correlation with the concentrations of $O_3$ and Ox in summer and sulfate, nitrate, and LWC in winter, although the correlation coefficients ($r^2$) with $O_3$ and $O_x$ were slightly decreased (by less than 0.1) while those with sulfate increased by 0.1-0.4 compared to the correlation analysis with the observed concentrations of OSs. This result indicates that the measurement uncertainties did not significantly influence the correlation analysis results and the evaluation of the dominant influencing factors for the formation of iSOA OSs in this study."

Table S7. Coefficients of correlation ($r^2$) between 2-MG-OS and 2-MT-OS and various influencing factors in summer and winter of 2015, 2019, and 2021.

| | | nitrate | sulfate | LWC | pH | NO$_2$ | O$_3$ | O$_x$ | isoprene | T |
|---|---|---|---|---|---|---|---|---|---|---|
| 2-MG-OS | Summer-2015 | 0.01 | 0.24 | 0 | 0.19 | 0.19 | 0.61 | 0.61 | 0.55 | 0.56 |
| | Summer-2019 | 0.19 | 0.32 | 0.02 | 0.01 | 0.53 | 0.42 | 0.51 | 0 | 0.27 |
| | Summer-2021 | 0.57 | 0.54 | 0.21 | 0.02 | 0.37 | 0.72 | 0.68 | 0.04 | 0.21 |
| | Winter-2015 | 0.75 | 0.64 | 0.47 | 0.04 | 0.32 | 0.23 | 0.1 | / | 0.02 |
| | Winter-2019 | 0.69 | 0.63 | 0.19 | 0.03 | 0.29 | 0.02 | 0.24 | 0.02 | 0.12 |
| | Winter-2021 | 0.6 | 0.71 | 0.68 | 0.04 | 0.21 | 0.04 | 0 | 0.15 | 0.03 |
| 2-MT-OS | Summer-2015 | 0.05 | 0.11 | 0.04 | 0.2 | 0.18 | 0.61 | 0.61 | 0.3 | 0.66 |
| | Summer-2019 | 0.11 | 0.24 | 0 | 0 | 0.52 | 0.33 | 0.42 | 0 | 0.27 |
| | Summer-2021 | 0.3 | 0.26 | 0.1 | 0.01 | 0.14 | 0.51 | 0.35 | 0.1 | 0.26 |
| | Winter-2015 | 0.8 | 0.6 | 0.48 | 0.04 | 0.41 | 0.26 | 0.14 | / | 0.05 |
| | Winter-2019 | 0.58 | 0.67 | 0.28 | 0.28 | 0 | 0.17 | 0.01 | 0.13 | 0.1 |
| | Winter-2021 | 0.65 | 0.79 | 0.64 | 0.06 | 0.13 | 0.03 | 0 | 0.07 | 0.02 |

e.     The current treatment of matrix effects is insufficient at lines 221-225. While the expected extent of matrix effects may be informative, correction factors are not valid across studies. The authors must discuss the relevance of the sample matrix in the study by Bryant et al. (2021) and their work. Similarly, it is not a valid approach to extrapolate relative ionization efficiencies observed by others (i.e. Bryant et al. (2021, line 169)) across studies, because ionization changes day-to-day within an instrument, and depends upon specific instrumental conditions and mobile phase composition. Ratios of relative responses are reported to 2-3 significant figures at lines 169-173, while in reality these estimates are known with much less certainty.

Response: We agree with the reviewer that the matrix effect factors and relative ionization efficiencies reported by Bryant et al. (2021) may not be valid for the present study, given that these factors could vary significantly across studies due to the differences in ambient samples analyzed and in analytical instruments and specific instrumental conditions. We have conducted additional experiments to evaluate the matrix effect of OS measurements using ambient samples in Shanghai and modified the relevant discussions (see our response to comment #1a).

In addition, we have removed the discussion about ratios of relative responses of OSs in lines 169-173 and added the following sentence to the manuscript.

Lines 162-163: "Use of surrogate standards would lead to uncertainties in measured concentrations of 2-MT-OS and 2-MG-OS (Bryant et al., 2021; Bryant et al., 2020), but not alter the inter-annual trend of iSOA OSs."

2. There are also opportunities for more thorough comparisons of the modeled and experimental data.  These are mentioned in the text, but should be integrated into figures. These additions would strengthen the conclusions of the paper and improve clarity.

a.     Extend Figure 2 be expanded to also show $PM_{2.5}$ mass, OC (or OM), sulfate, nitrate, and other relevant PM component or atmospheric parameters ($NO_x$, $O_3$). This would be a useful way to provide context for understanding changes in isoprene SOA that are discussed subsequently.

Response: Thanks. The relevant PM components and atmospheric parameters have been added in Figure 2.

[Figure]

Figure 2. Seasonal and inter-annual variations concentration of PM$_{2.5}$ and its major components (a-e), gas-phase anthropogenic pollutants (f-h), as well as particulate iSOA tracers, including (i)2-MTs, (j) 2-MT-OS, (k) C$_5$-alkene triols, (l) IEPOX-SOA (the sum of 2-MTs, 2-MT-OS, and C$_5$-alkene triols), (m) 2-MG, (n) 2-MG-OS, (o) HMML&MAE-SOA (2-MG plus 2-MG-OS), and (p) iSOA (the sum of all tracers).

b.    In figures 6 and 7, include experimental data be for comparison to modeled values.

Response: Thanks. The observed data have been added in Figure 6 and 7 as follows:

[Figure]

Figure 6. Simulated concentrations of 2-MTs, 2-MT-OS, 2-MG and 2-MG-OS in summer (a-d) and winter (e-h) in Case 1 (2015-sim and 2019-sim) and Test Case (simulations with 2015 emissions and 2019 meteorological conditions). The observed concentrations of 2-MTs, 2-MT-OS, 2-MG, and 2-MG-OS in 2015 and 2019 are also displayed (Detailed model-measurement comparisons are provided in Section 3.2; after accounting for matrix effect, 2-MT-OS would decrease more sharply and 2-MG-OS would show a descending inter-annual trend, consistent with model simulations).

[Figure]

Figure 7. Simulated EC concentrations in Case 1 (2015-sim and 2019-sim) and Test Case (simulations with 2015 emissions and 2019 meteorological conditions) in (a) summer and (b) winter. The measured concentrations are also displayed and their trend is consistent with the simulations.

c.    Also in figures 6 and 7 – the x-axis labels are confusing. Can they be simplified?

Response: Thanks. We have revised the x-axis labels in Figures 6 and 7 (see above).

3. In many places, the authors compare the current study to select literature references. Many of these comparisons are to background or rural locations (i.e. central Amazonia and rural sites in the Southeastern United States), raising question as to their relevance to Shanghai. To better understand the urban influence (and the emissions reductions) on isoprene SOA chemistry, the authors should more thoroughly compare and discuss their work in relation to prior studies in urban locations in Asia and elsewhere.

Response: Thanks for the reviewer's suggestion. We have thoroughly compared our measurements to prior studies in urban regions (see Table S6) and revised the discussion in the manuscript accordingly. The revised text of the relative abundance of iSOA OS tracers to polyol tracers in our work compared to other measurements is provided in our response to comment #1c, and other results in our work, including dominance of IEPOX-SOA over HMML&MAE-SOA and correlation relationship between 2-MG and ozone, has been compared to previous measurements (see below).

Lines 404-407: "The dominance of IEPOX-SOA over HMML&MAE-SOA in summer is in agreement with RPLC-MS measurements in Beijing, Hefei and Kunming in China (Zhang et al., 2022b) and Birmingham, US (Rattanavaraha et al., 2016), as well as hydrophilic interaction liquid chromatography-mass spectrometry (HILIC-MS) measurements conducted in urban Guangzhou, China (Liu et al., 2025)."

Lines 485-486: "Such correlations between 2-MG and ozone were also observed in previous measurements in urban areas in southeastern US (Rattanavaraha et al., 2016)…"

Table S6. Concentrations (ng m$^{-3}$) of 2-MT-OS, 2-MTs, 2-MG-OS and 2-MG measured in different urban environments

| Sampling sites | Sampling time | OS detection method | 2-MT-OS | 2-MTs | 2-MG-OS | 2-MG | Refs. |
|---|---|---|---|---|---|---|---|
| Urban Beijing, China | Summer, 2014 | RPLC/ESI-MS | 0.5 [a] | 7.52 [b] | 0.99 [a] | 2.76 [b] | Zhang et al. (2022a) |
| Urban Hefei, China | Summer, 2014 | RPLC/ESI-MS | 0.71 [a] | 17.7 [b] | 1.68 [a] | 4.37 [b] | Zhang et al. (2022a) |
| Urban Kunming, China | Summer, 2014 | RPLC/ESI-MS | 1.43 [a] | 25.1 [b] | 1.49 [a] | 3.26 [b] | Zhang et al. (2022a) |
| Urban Birmingham, Alabama, USA | Summer, 2013 | RPLC/ESI-MS | 165 [c] | 374 [d] | 7.20 [c] | 10.4 [d] | Rattanavaraha et al. (2016) |
| Urban Beijing, China | Spring, 2017 | RPLC/ESI-MS | 11.8 [d] | 17.3 [b] | 21 .5 [d] | 7.2 [b] | Bryant et al. (2020) |
| Urban San Agustin, Mexico | Summer, 2018 | HILIC/ESI-MS | 20 [d] | 40 [d] | / | / | Cooke et al. (2024) |
| Urban Manaus, Brazil | Winter, 2016 | HILIC/ESI-MS | 0.39 [d] | 0.14 [d] | / | / | Cui et al. (2018) |
| Urban Guangzhou, China | Summer, 2018 | HILIC/ESI-MS | 62.8 [e] | 60.5 [b] | 13.8 [e] | 2.9 [b] | Liu et al. (2025) |
| Urban Guangzhou, China | Fall, 2018 | HILIC/ESI-MS | 29.7 [e] | 16.6 [b] | 7.7 [e] | 3.0 [b] | Liu et al. (2025) |

Note: Concentrations of target compounds quantified by: [a]camphorsulfonate, [b]erythritol, [c]propyl sulfate, [d]authentic standard, and [e]ethyl sulfate.

4. When comparing to prior studies, the authors need to consider how similar or different methods may influence the comparison.

Response: Thanks for the reviewer's suggestion. The type of chromatographic columns used for quantification in prior studies are summarized in Table S6. When comparing the result in this work and previous measurements, the influence of similar or different methods have been considered (see response to #1c and #3).

5. The notation "HMML/MAE" implies the ratio of HHML / MAE. Is this intended? Or could this be HHML, MAE, or HHML & MAE?

Response: The notation "HMML/MAE" indicates the NOx-dominant pathway with HMML or MAE as the reaction intermediates. We have replaced it with "HMML & MAE" in the revised manuscript.

6. At least one of the studies mentioned at line 166 (Hettiyadura et al. 2015) did not use camphorsulfonic acid as a surrogate standard. Please check and revise this thoroughly.

Response: Thanks for pointing out this. We have deleted this sentence and added the following statement regarding the use of surrogate standards in the revised manuscript.

Lines 162-163: "Use of surrogate standards would lead to uncertainties in measured concentrations of 2-MT-OS and 2-MG-OS (Bryant et al., 2021; Bryant et al., 2020), but not alter the inter-annual trend of iSOA OSs."

7. At line 38, it is it a realistic recommendation to "regulate atmospheric oxidizing capacity"? Typically, regulations are either on emission sources or on ambient concentrations of hazardous pollutants. Please reconsider this closing statement in the abstract.

Response: Thanks for the reviewer's comment. We have modified this sentence as "These findings highlight pathway-specific iSOA responses to emission reductions in a megacity and the importance of targeted anthropogenic emission reductions for mitigating biogenic SOA formation through regulating atmospheric oxidizing capacity and aerosol reactivity."

8. Line 67, a symbol before 45 is appearing as a box.

Response: Thanks. We have modified this.

9. The logic at lines 98-104 needs improvement. The approach that "previous studies have mainly focused on the characterization of the particle-phase abundance of the iSOA tracers" is justified in that for iSOA to form it must be in the particle phase. Additionally, this paragraph implies that gas phase concentrations of tracers were measured in this study, whereas they appear only to be estimated by calculation and not measured.

Response: Thanks for the reviewer's suggestion. We have modified the text for a clearer logic:

Lines 98-102: "Furthermore, while the iSOA polyol tracers are formed in the particle phase, they can actively partition between gas and particle phases due to their semi-volatile characteristics (Fan et al., 2020; Isaacman-Vanwertz et al., 2016; Nguyen et al., 2015). As a result, considering their particle-phase concentration only may bias our understanding of the atmospheric abundance and chemistry of iSOA."

10. Can the authors validate their estimates of gas-particle distributions using experimental data?

Response: Thanks for the reviewer's suggestion. However, currently we are not able to perform such validations due to the lack of the measured concentration of iSOA polyol tracers in the gas phase. Previous observational studies have determined a gas-phase fraction of 40-80% for 2-MG (Yee et al., 2020; Nguyen et al., 2015) and approximately 50% for 2-MTs (Yee et al., 2020; Isaacman-Vanwertz et al., 2016). Moreover, the gas-phase fraction of 2-MG exhibited a negative correlation with pH when partitioning between gas phase and bulk solution (Nguyen et al., 2015) or ambient $PM_{2.5}$ (Yee et al., 2020). When pH was lower than 3, over 50% of 2-MG partitioned into the gas phase in southeastern U.S. and central Amazon (Yee et al., 2020). In this study, the pH values ranged from 2 to 4, suggesting a substantial fraction of 2-MG in the gas phase.

11. Table S2, it appears that matrix effects were considered in only six samples, not eight as suggested by the caption.

Response: Thanks. The caption has been revised.

12. In Figure 1, is wind speed data missing from 2021 and 2022? It seems not to appear in the figure shown. In general, the resolution and the quality of this figure should be improved prior to publication.

Response: Yes, the wind speed data were not collected for 2021. We have added a statement on this to the figure caption. We have also replaced Figure 1 with a version of higher resolution.

[revised manuscript text omitted]

Zhang, Y.-Q., Ding, X., He, Q.-F., Wen, T.-X., Wang, J.-Q., Yang, K., Jiang, H., Cheng, Q., Liu, P., Wang, Z.-R., He, Y.-F., Hu, W.-W., Wang, Q.-Y., Xin, J.-Y., Wang, Y.-S., and Wang, X.-M.: Observational Insights into Isoprene Secondary Organic Aerosol Formation through the Epoxide Pathway at Three

Urban Sites from Northern to Southern China, Environ. Sci. Technol., 10.1021/acs.est.1c06974, 2022a.

Zhang, Y.-Q., Ding, X., He, Q.-F., Wen, T.-X., Wang, J.-Q., Yang, K., Jiang, H., Cheng, Q., Liu, P., Wang, Z.-R., He, Y.-F., Hu, W.-W., Wang, Q.-Y., Xin, J.-Y., Wang, Y.-S., and Wang, X.-M.: Observational Insights into Isoprene Secondary Organic Aerosol Formation through the Epoxide Pathway at Three Urban Sites from Northern to Southern China, Environ. Sci. Technol., 10.1021/acs.est.1c06974, 2022b.

We are grateful to the reviewer for the thoughtful comments, which are very helpful for improving our manuscript. Our point-to-point responses to each comment are as follows (the reviewer's comments are in black text, our responses are in blue text, and revised texts that appear in the manuscript are in red text).

General Comments:

The manuscript by Hu et al. presents interesting new results on isoprene-derived SOA in a megacity in China and the response to the on-going reductions in anthropogenic emissions. In general, the results are well presented and discussed. The study includes both measurements and modelling results. One limitation, which the authors should keep in mind, is that the field measurements are conducted on a limited number of samples. The modelling approach balances this, but the text sometimes overstates the conclusions that can be drawn.

Specific comments:

Line 21: profoundly is a somewhat strange word to use here.

**Response**: We have replaced "profoundly" with "strongly".

L28: Mass spectrometry – please be more specific.

**Response:** We have specified the type of mass spectrometry in the text.

L26-27: "…were measured by gas chromatography-mass spectrometry and high-resolution liquid chromatography-mass spectrometry."

L38: "regulating atmospheric oxidizing capacity and aerosol reactivity to mitigate biogenic SOA formation" Is biogenic SOA formation really the major source of air pollution? How can atmospheric oxidizing capacity and aerosol reactivity be regulated?

Response: Thanks for the reviewer's comment. The contribution of biogenic SOA to ambient particulate matter varies significantly across different locations and seasons. For example, observational studies revealed that biogenic SOA contributed to approximately half of total fine organic aerosol in southeastern US (Liao et al., 2007; Zhang et al., 2018). Modelling studies found that biogenic sources accounted for 61% of SOA on average in China, with its proportion reaching 75% in summer but dropping to 24% in winter (Hu et al., 2017). In addition, the proportion of biogenic SOA increased with anthropogenic emission reductions in recent years (Zhang et al., 2024). These results suggest an important contribution of biogenic SOA to PM pollution.

The atmospheric oxidizing capacity is largely governed by the concentrations of oxidants including hydroxyl radicals, ozone, and nitrate radicals, which can be regulated by reducing anthropogenic emissions such as VOCs and NOx (Fiore et al., 2024; Dai et al., 2024). The aerosol reactivity is controlled by its concentration and composition, which can be regulated by reducing anthropogenic emissions such as VOCs, NOx, and $SO_2$.

In the revised manuscript, this sentence has been revised as follows:

Line 36-39: "These findings highlight pathway-specific iSOA responses to emission reductions in a megacity and the importance of targeted anthropogenic emission reductions for mitigating biogenic SOA formation through regulating atmospheric oxidizing capacity and aerosol reactivity."

L46: Heald et al., 2008 -is there a newer reference.

Response: Thanks. The reference has been updated here as follows:

L44-46: "…the oxidation of isoprene contributes significantly to the formation of secondary organic aerosol (SOA) in the troposphere, estimated to 19.6 TgC $yr^{-1}$ (Kelly et al., 2018)."

L58-59: "The gaseous IEPOX can be taken up into aqueous aerosol and undergo acid-catalyzed reactions to form polyols, organosulfates (OSs), and oligomers." This sentence needs references.

Response: Thanks. We have added references to this sentence.

L57-60: "The gaseous IEPOX can be taken up into aqueous aerosol and undergo acid-catalyzed reactions to form polyols, organosulfates (OSs), and oligomers (Hettiyadura et al., 2019; Lin et al., 2012; Surratt et al., 2010)."

L61: "abundantly measured" it is unclear what you mean – where they measured a lot or in high concentrations?

Response: We want to mean here the IEPOX-SOA tracers were measured a lot. To be more precise, we have replaced "abundantly" with "extensively".

L67: The sign before 45% is missing.

Response: Thanks. The sign before 45% is "~", and we have added it.

L100: Which iSOA tracers?

Response: The iSOA tracers here include polyols tracers such as 2-MTs and 2-MG. We have replaced the sentence here with a clearer statement.

Lines 98-102: "Furthermore, while the iSOA polyol tracers are formed in the particle phase, they can actively partition between gas and particle phases due to their semi-volatile characteristics (Fan et al., 2020; Isaacman-Vanwertz et al., 2016; Nguyen et al., 2015). As a result, considering their particle-phase concentration only may bias our understanding of the atmospheric abundance and chemistry of iSOA."

L108: cation ions -> cations

Response: We have revised this.

L109: The text is about recent changes but one of the references is from 2014.

Response: We have updated the reference to newer ones.

L104-107: "For example, $PM_{2.5}$ mass concentrations dropped significantly, with a marked reduction in water-soluble inorganic ions such as sulfate, ammonium, and cation ions (Liu et al., 2023a; Zheng et al., 2018; Liu et al., 2023b; Geng et al., 2024)…"

L113: later -> latter

Response: We have corrected this typo.

L119: comprehend -> understand

Response: We have revised this.

L149. What is KPA?

Response: KPA refers to ketopinic acid that has been widely used as a surrogate standard in the quantification of iSOA polyol tracers. We have replaced KPA with its full name ketopinic acid.

L151. Please provide information about sources of authentic standards.

Response: Thanks. We have added the sources of the standards.

L149-150: "…2-MG and 2-methylerythritol were quantified using their authentic standards (Toronto Research Chemicals, 99.8%)."

L171-172: Can this depend on the vendor of the ESI inlet and MS?

Response: Yes, the ESI efficiency is affected by various factors, including the type and configuration of the ESI inlet and MS as well as their specific operating conditions. Considering that the ESI efficiency obtained from one instrument may not be directly applied to another instrument, we have deleted the

discussion in lines 171-172.

L181: Please be a bit more specific about the distance and the surrounding areas i.e. emissions.

Response: We have added a more detailed description about this.

L169-173: "Temperature, relative humidity (RH), as well as the concentrations of trace gases and $PM_{2.5}$ were measured at a state-controlled air quality monitoring station on the Xuhui Campus of Shanghai Normal University, which is surrounded by residential areas and commercial districts and 4.5 km southwest of the $PM_{2.5}$ sampling site of this work."

L194-195: Please discuss the uncertainty associated with this approach.

Response: Previous studies have found that lacking gas-phase inputs of ammonia could lead to underprediction of pH using thermodynamic equilibrium models, such as ISORROPIA and E-AIM (Hennigan et al., 2015; Guo et al., 2015; Song et al., 2018). Guo et al. (2015) found that the pH values were underestimated by 1 unit on average in southeast U.S. when using only aerosol ammonium data as inputs in ISORROPIA model. Similarly, Song et al. (2018) found that a 10-fold increase in gas-phase $NH_3$ concentrations roughly corresponds to a 1 unit increase in pH in the ammonia-rich atmosphere like Beijing. In addition, we found that the average difference between aerosol pH predicted by using aerosol ammonium only and gas-plus-particle-phase ammonia as inputs was about 1 unit in 2015 and 2019. Thus, we inferred that the lack of gas-phase concentrations of ammonia might lead to underestimation of pH by ~1 unit and increased the output pH estimated using aerosol ammonium only as input by one unit in 2021 to reduce the uncertainty arising from lack of gas-phase $NH_3$ in pH estimation.

We have added the following discussion to the revised manuscript.

L182-195: "Additionally, the 2021 pH was predicted by the ISORROPIA-II using particle-phase-only concentrations of ions as input due to lack of gas-phase $NH_3$ data. Previous studies have found that lacking gas-phase inputs of ammonia could lead to under-prediction of pH using thermodynamic equilibrium models, such as ISORROPIA and E-AIM (Hennigan et al., 2015a; Guo et al., 2015; Song et al., 2018). Guo et al. (2015) found that the pH values were underestimated by 1 unit on average in southeast US when using only aerosol ammonium data as inputs in ISORROPIA model. Similarly, Song et al. (2018) found that a 10-fold increase in gas-phase $NH_3$ concentrations roughly corresponds to a 1 unit increase in pH in the ammonia-rich atmosphere like Beijing. In addition, we found that aerosol pH in 2015 and 2019 predicted using aerosol ammonium only as input in our study was on average 1 unit lower than that predicted using gas-plus-particle-phase ammonia as input in Zhou et al. (2022). Thus, we inferred that the lack of gas-phase concentrations of ammonia might lead to underestimation of pH by ~1 unit in the present study and increased the output pH estimated using aerosol ammonium only as input by one unit to represent aerosol acidity in 2021."

L224: It is nice that the authors bring forward this uncertainty, but it is unclear how it affects the results presented here and whether it should have been tested if the magnitude of the effect is the same in the current study.

Response: Thanks for the reviewer's comment. Reviewer #1 has also raised this concern (comment #1) and we have addressed it in detail in our responses to his/her comments.

The matrix effect factors and relative ionization efficiencies reported by Bryant et al. (2021) may not be valid for the present study, given that these factors could vary significantly across studies due to the differences in ambient samples analyzed and in analytical instruments and specific instrumental conditions. Therefore, we have conducted a set of experiments to evaluate the matrix effect of iSOA OSs and the uncertainties in the abundance, trend, and relative ratios of different iSOA tracers in the present study and added relevant discussions to the revised manuscript (see below).

Lines 221-255: "Previous studies have demonstrated that the concentrations of 2-MT-OS and 2-MG-OS were significantly underestimated due to matrix effect by using reversed phase liquid chromatography-mass spectrometry (RPLC-MS) (Hettiyadura et al., 2015; Bryant et al., 2020; Bryant et al., 2021). In the present work, because of lack of authentic standards of isoprene-derived OSs, we are not able to quantify the absolute value of underestimation in the concentration of 2-MT-OS and 2-MG-OS due to the matrix effect. However, using ambient $PM_{2.5}$ samples with different concentrations, we can quantify the relative extent of underestimation in OS concentrations due to matrix effect in different samples, which allows for an evaluation of uncertainties in the abundance, trend, and relative ratios of different iSOA tracers in

this study. In the matrix effect experiments, the extracts of ambient $PM_{2.5}$ samples with different concentrations were mixed and the measured signals of 2-MT-OS and 2-MG-OS in mixed extracts were compared to the sum of OS signals detected separately in individual extracts. The concentrations of $PM_{2.5}$, sulfate, as well as 2-MT-OS and 2-MG-OS in ambient samples used for this evaluation are listed in Table S3. The relative matrix effect factor ($F_{matrix}$), defined as the ratio of the measured OS signals in mixed extracts to the sum of OS signals measured in each extract before mixing, are used to evaluate the matrix effect of OSs. A $F_{matrix}$ value of less than 1 indicates the presence of matrix effect.

As shown in Figure S2, the $F_{matrix}$ values were significantly smaller than 1 in both summer and winter, indicating that the signal responses of 2-MT-OS and 2-MG-OS in mixed extracts were largely suppressed due to the matrix effect. Notably, $F_{matrix}$ exhibits a significant negative dependence on the reduced mass ($\mu$, $\mu g\ m^{-3}$), a proxy used to represent effective mass loadings of the mixed $PM_{2.5}$ extracts, defined as:

$$\mu = \sqrt{m_1 * m_2 / (m_1 + m_2)} \qquad (1)$$

where $m_1$ and $m_2$ are the $PM_{2.5}$ mass loading of individual samples. This observation suggests that the concentrations of iSOA OSs in $PM_{2.5}$ samples collected in 2015 were underestimated more than those in 2021, given that ambient $PM_{2.5}$ concentrations declined by 39.8% and 47.0% from 2015 to 2021 in summer and winter, respectively.

As $PM_{2.5}$ concentrations in ambient samples used for the matrix effect evaluation generally represent lower or upper ends of $PM_{2.5}$ concentrations during the observation period (see Table S3), the relative differences in measured $F_{matrix}$ values at varying reduced mass (Figure S2) may roughly reflect the differences in the extent of underestimation in OS concentrations for samples collected across 2015-2021. During summer, the $F_{matrix}$ value decreased from 0.71 to 0.63 for 2-MT-OS and from 0.85 to 0.58 for 2-MG-OS with increasing reduced mass, indicating that the concentrations of these two iSOA OSs were a factor of 1.2 and 1.5 more underestimated in 2015 than in 2021 due to matrix effect. Similarly, during winter the $F_{matrix}$ values of iSOA OSs decreased from 0.9 to 0.6 with rising reduced mass, implying a factor of 1.5 greater underestimation in OS concentrations in 2015 than in 2021."

Lines 371-379: "However, the inter-annual trend of iSOA OSs could be altered due to the matrix effect. The measured concentration of 2-MT-OS exhibited a decreasing inter-annual trend, while 2-MG-OS showed insignificant variation between 2015-2021. Accounting for the significantly larger matrix effects in 2015 samples compared to 2021 samples (see Section 2.3), the true concentrations of 2-MT-OS would decrease more sharply and 2-MG-OS might also exhibit a declining trend."

Lines 411-413: "However, given the significant underestimation of iSOA OSs due to matrix effect and overestimation of $C_5$-alkene triols due to their potential artifact formation, the true concentration of iSOA OSs would predominate over that of polyol tracers."

Table S3. Major components in eight $PM_{2.5}$ filter samples that used for estimating matrix effect of 2-MT-OS and 2-MG-OS.

| Season | Sample | Sampling date | $PM_{2.5}$ ($\mu g\ m^{-3}$) | $SO_4^{2-}$ ($\mu g\ m^{-3}$) | 2-MG-OS ($ng\ m^{-3}$) | 2-MT-OS ($ng\ m^{-3}$) |
|---|---|---|---|---|---|---|
| Summer | High 1 | 2015/7/28 | 35.66 | 7.74 | 13.04 | 157.89 |
| | High 2 | 2019/7/24 | 31.76 | 4.58 | 12.36 | 100.61 |
| | Low 1 | 2019/8/5 | 10.04 | 2.29 | 0.58 | 0.83 |
| | Low 2 | 2021/8/5 | 12.21 | 1.48 | 0.18 | 0.18 |
| Winter | High 3 | 2022/1/2 | 77.76 | 6.28 | 1.80 | 0.47 |
| | High 4 | 2022/1/3 | 65.38 | 5.41 | 1.54 | 0.41 |
| | Low 3 | 2021/12/25 | 18.04 | 3.42 | 0.49 | 0.07 |
| | Low 4 | 2021/12/26 | 23.71 | 3.01 | 0.70 | 0.08 |

[Figure]

Figure S2. Relative matrix effect factors (defined as the ratio of signal response in mixed extracts to the sum of signal response in individual extracts, $F_{matrix}$) of 2-MT-OS and 2-MG-OS under different mixture type in summer (a) and winter (b) (yellow background represents low-plus-low mixture, green background represents high-plus-low mixture, and blue background represents high-plus-high mixture).

L299: Are the values statistically significantly different?

Response: Results from a significance test demonstrated a statistically significant difference in $O_3$ concentrations between 2015 and 2019, but not between 2019 and 2021.

The description here has been revised as follows:

Line 327-329: "By contrast, the concentration of $O_3$ significantly decreased from $52.0 \pm 38.9$ ppb in 2015 to $41.2 \pm 22.8$ ppb in 2019 ($p < 0.05$) and then remained at a comparable level ($43.4 \pm 20.8$ ppb) in 2021…"

L302: Dramatic is not a correct scientific word to use here. It would be nice if the authors could list some numbers.

Response: Thanks. The word "dramatic" was replaced by "drastic", and the proportion of major composition of $PM_{2.5}$ have been added there.

L330-332: "During the observation period, the average $PM_{2.5}$ concentration decreased by 41.7% from 2015 to 2021, with concentrations of major components, including sulfate, ammonium, and OM, decreasing by 51.8%, 40.6%, and 39.1%, respectively (Figure 2)."

L303: Are there other types of nitrate than aerosol nitrate?

Response: There are no other types of nitrate than aerosol nitrate. For simplicity, we have replaced "aerosol nitrate" with "nitrate".

L305: Please also state the average values.

Response: Thanks, we have added the average values for the fraction.

L335-337: "Overall, OM was the most abundant component in $PM_{2.5}$, accounting for 10.2-72.7% (average 22.6%) of total $PM_{2.5}$ mass, followed by sulfate (6.8-45.2%, average 17.7%), nitrate (0.5-32.6%,

average 17.0%), and ammonium (1.1-18.2%, average 8.8%)."

L308: Please avoid the use of the word dramatic. Aerosol pH – are the differences statistically significantly different?

Response: Thanks. We have conducted a t-test for aerosol pH and LWC, and the result shows that the differences in aerosol pH and LWC across different years are significantly different ($p < 0.05$).

We have modified the sentence as follows:

L337-340: "Ascribed to the strong decrease of inorganic ion concentrations (in particular sulfate), aerosol LWC decreased from $9.14 \pm 4.51$ µg m$^{-3}$ in 2015 to $4.40 \pm 2.76$ µg m$^{-3}$ in 2021 ($p < 0.05$). Aerosol pH decreased from $3.2 \pm 0.4$ in 2015 to $2.5 \pm 0.9$ in 2021 ($p < 0.05$)…"

Figure 1: The figure is very small and it is difficult to see details.

Response: We have updated Figure 1 to a clearer version.

[Figure]

L334: Please state standard deviations.

Response: Thanks. We have included the standard deviations for the concentration values.

L408-409: "The annual average particulate concentration of the total iSOA polyol tracers (including 2-MTs, 2-MG, and C$_5$-alkene triols) were $36.1 \pm 63.3$, $33.4 \pm 70.5$, and $18.7 \pm 47.1$ ng m$^{-3}$ in 2015, 2019 and 2021…"

L361: What do you mean by median?

Response: We have replaced "median" with "moderate".

L365-367: Please clarify this sentence.

Response: We have deleted this sentence since we have added detailed discussion about the modeled result of individual iSOA tracers in the revised manuscript.

L368: Is it consistent or different?

Response: Thanks. There is a typo here and we have deleted the word "in" to emphasize the consistence between the model simulation results in our study and previous studies.

L396-398: It seems like a bold statement to say that the concentrations in winter decrease when the values are $0.8 \pm 0.3$, $0.8 \pm 0.2$, and $0.7 \pm 0.4$.

Response: We have deleted this statement since the large uncertainties arising from thermal degradation of $C_5$-alkene triols and the matrix effect of iSOA OSs would make it difficult to discuss the inter-annual trend of the IEPOX-SOA/HMML&MAE-SOA ratio.

L400-403: These sentences need clarification.

Response: Thanks. These sentences have been deleted since large uncertainties existing in quantification of $C_5$-alkene triols makes it difficult to discuss the trend of the ratios of IEPOX-SOA/HMML&MAE-SOA.

Figure 4: It is not possible to read the numbers in the heatmap.

Response: Thanks. The heatmap has been replaced with a clearer version.

[Figure]

Figure 4. Coefficients of correlation ($r^2$) between iSOA compounds and various influencing factors of iSOA formation in (a-c) summer and (d-f) winter of 2015, 2019, and 2021, respectively. The ISOP is the abbreviation of isoprene.

L429-431: The correlation is quite low, which should be reflected better in the text.

Response: We have used a more neutral description for the correlation, replacing "moderate" with "weak to moderate".

L434-436: The correlation is quite low to moderate, which should be reflected better in the text.

Response: We have replaced "well" with "moderately".

L439: The change is quite small. Was a statistical test performed to check this?

Response: A t-test was performed to evaluate the statistical significance of the changes in wintertime IEPOX-SOA, LWC, and sulfate concentrations across different years. The result shows that the changes are statistically significant ($p < 0.05$) during the observation period.

L451: What happens to isoprene when the oxidation capacity is lower? Does it not just take longer for the oxidation to occur?

Response: Yes, the lower oxidation capacity generally means lower level of oxidants, thus the oxidation of isoprene would be slower, which could lead to decreased formation of iSOA.

L496-498: Unclear sentence.

Response: The sentence has been rephrased as follows:

L564-566: "It also suggests that the $k_{i,j\text{-IEPOX}}^{-4}$ is more appropriate for predicting aerosol heterogeneous reactivity toward IEPOX than other four sets of kinetic parameters."

Figure 6: Describe better what the numbers/years for the cases are.

Response: The x-axis label in Figure 6 has been revised and its meaning is explained in the figure caption. In addition, we have added the measured data to this figure according to Reviewer #1's suggestion.

[Figure]

Figure 6. Simulated concentrations of 2-MTs, 2-MT-OS, 2-MG and 2-MG-OS in summer (a-d) and winter (e-h) in Case 1 (2015-sim and 2019-sim) and Test Case (simulations with 2015 emissions and 2019 meteorological conditions). The observed concentrations of 2-MTs, 2-MT-OS, 2-MG, and 2-MG-OS in 2015 and 2019 are also displayed (Detailed model-measurement comparisons are provided in Section 3.2; after accounting for matrix effect, 2-MT-OS would decrease more sharply and 2-MG-OS would show a descending inter-annual trend, consistent with model simulations).

L551: This line needs editing.

Response: Thanks. This sentence has been revised as follows:

L626-630: "On the other hand, MACR is known as a first-generation oxidation product of isoprene and an important intermediate for iSOA formation through NOx-dominant pathways (Nguyen et al., 2015), but it can also originate from primary sources, such as biological emissions (Jardine et al., 2012), residential wood burning (Gaeggeler et al., 2008), and vehicle exhaust emissions (He et al., 2009)"

L554: "Field studies have demonstrated" – this is a bold statement as there are differences between urban areas.

Response: This statement has been revised to improve clarity.

L630-632: "In certain urban areas, MACR arises primarily from vehicular emissions, as observed at the Heshan site in the Pearl River Delta region, China (Ling et al., 2019a) and in Houston, US (Park et al., 2011). "

L564: underappreciated is a loaded word not suitable here.

Response: We have replaced the word "underappreciated" with "underpredicted".

L567: three- year measurement gives the impression that samples for three complete years were studied. Please correct.

Response: We have replaced "a three-year measurement" with "an observation" and added "during the summer and winter in 2015, 2019, and 2021" to the end of this sentence to more accurately reflect the temporal coverage of the data.

L568: During the period 2015-2021, however only periods during three years were studied. Please correct.

Response: We have replaced "during the period 2015-2021" with "during the summer and winter in 2015, 2019, and 2021."

L574-575: "while HMML/MAE-SOA species (4.3, 4.2, and 4.3 ng m$^{-3}$ in 2015, 2019, and 2021, respectively) did not decrease significantly". No they did not seem to decrease at all! Please correct.

Response: Thanks. We have deleted the word "significantly" in this sentence.

L579-581: Please check that this can be concluded based on the current study.

Response: Thanks. The correlation analysis showed that iSOA exhibited a good correlation with O$_3$ and Ox in summer and sulfate in winter. This indicates that the level of atmospheric oxidants (that largely determines the rate of isoprene oxidation) and sulfate aerosol (that plays a crucial role in the reactive uptake of isoprene-derived epoxide intermediates) are key factors driving iSOA formation in summer and winter, respectively. Based on this, we can conclude that "The atmospheric oxidation of isoprene to epoxide intermediates and their subsequent reactive uptake on aqueous aerosols are the key steps driving the formation of iSOA in summer and winter, respectively."

L582-583: This sentence is not clear.

Response: This sentence has been revised as follows:

L657-658: "The Ox-represented atmospheric oxidizing capacity and aerosol heterogeneous reactivity decreased significantly during the observation period …"

L599: Are biogenic emissions pollution?

Response: We have modified "PM pollution from biogenic emissions" as "PM formation from biogenic emissions".